# Provably Efficient Offline Multi-agent Reinforcement Learning via Strategy-wise Bonus

**Qiwen Cui**
Paul G. Allen School of Computer Science
Engineering
University of Washington
qwcui@cs.washington.edu

**Simon S. Du**
Paul G. Allen School of Computer Science
Engineering
University of Washington
ssdu@cs.washington.edu

## Abstract

This paper considers offline multi-agent reinforcement learning. We propose the strategy-wise concentration principle which directly builds a confidence interval for the joint strategy, in contrast to the point-wise concentration principle that builds a confidence interval for each point in the joint action space. For two-player zero-sum Markov games, by exploiting the convexity of the strategy-wise bonus, we propose a computationally efficient algorithm whose sample complexity enjoys a better dependency on the number of actions than the prior methods based on the point-wise bonus. Furthermore, for offline multi-agent general-sum Markov games, based on the strategy-wise bonus and a novel surrogate function, we give the first algorithm whose sample complexity only scales $\sum_{i=1}^{m} A_i$ where $A_i$ is the action size of the $i$-th player and $m$ is the number of players. In sharp contrast, the sample complexity of methods based on the point-wise bonus would scale with the size of the joint action space $\Pi_{i=1}^{m} A_i$ due to the curse of multiagents. Lastly, all of our algorithms can naturally take a pre-specified strategy class $\Pi$ as input and output a strategy that is close to the best strategy in $\Pi$. In this setting, the sample complexity only scales with $\log |\Pi|$ instead of $\sum_{i=1}^{m} A_i$.

## 1 Introduction

Multi-agent reinforcement learning (MARL) is about decision making in a multi-agent system under uncertainty, which has achieved significant success in solving a wide range of tasks such as GO [Silver et al., 2017], Poker [Brown and Sandholm, 2019] and autonomous deriving [Shalev-Shwartz et al., 2016]. One standard setting in MARL is multi-player general-sum Markov games where each player deploys a policy to maximize its own total reward while the evolution of the environment depends on the policies of all the players [Zhang et al., 2021a]. During the learning process, each player needs to identify the environment dynamics as well as compete/cooperate with other agents.

One emerging subarea is offline MARL, where plenty of empirical works have been done while the theoretical understanding is still largely missing [Pan et al., 2021, Jiang and Lu, 2021, Meng et al., 2021]. Offline RL has received tremendous attention because in various practical scenarios, it is expensive to acquire online data while offline log data is accessible.

The offline single-agent RL is well studied in the literature. Researchers have identified the minimal dataset coverage assumption, *single policy coverage* (the dataset only needs to cover an optimal policy), under which one can learn a near-optimal policy efficiently. Furthermore, they have developed algorithms with minimax sample complexity [Xie et al., 2021b, Li et al., 2022]. For offline MARL, recent works showed that single policy coverage is not sufficient and *unilateral coverage* is necessary for learning a Nash equilibrium (NE) strategy, i.e., the dataset covers all the joint strategies that only differ from an NE at one player [Cui and Du, 2022, Zhong et al., 2022]. This condition is also

sufficient for two-player zero-sum Markov games with sample complexity $\widetilde{O}(AB)$ (ignoring other quantities), where $A$, $B$ are the number of actions for each player [Cui and Du, 2022]. However, it is still unclear if it is sufficient for multi-player general-sum Markov game.

One major challenge in MARL is the *curse of multiagents* [Jin et al., 2021a]. Suppose the number of actions for player $j$ is $A_j$ and there are $m$ players. Then the joint action space is of size $\prod_{j \in [m]} A_j$, which grows exponentially with the number of players $m$. As a result, any algorithm that depends linearly on the cardinality of the joint action space can hardly be applied to real-world scenarios. In online MARL, Jin et al. [2021a] and Song et al. [2021] show that finding the coarse correlated equilibrium, which is a weaker equilibrium notion than NE, only requires $\widetilde{O}(\max_{j \in [m]} A_j)$ samples, thus breaking the curse of multiagents. In this paper, we study the following question:

*Can we find NE in offline $m$-player general-sum Markov game with unilateral coverage and without the exponential dependence on the number of players?*

In this paper, we answer this question in the affirmative. We highlight our contributions below.

## 1.1 Main Novelties and Contributions

**1. Strategy-wise concentration principle.** We propose the strategy-wise concentration principle. Point-wise concentration is a standard technique in computing the confidence interval for each state-action pair [Azar et al., 2017, Liu et al., 2021, Xie et al., 2021b, Cui and Du, 2022]. However, the straightforward extension to MARL suffers from the curse of multiagents as the NE can be a mixed strategy. Different from the point-wise concentration technique, strategy-wise concentration directly *estimates each strategy, which allows a tighter confidence interval that can avoid the dependence on the joint action space.* We give a technical overview in Section 1.2. In addition, we show that the strategy-wise confidence bound is always a convex function so that the empirical *best response strategy can always be a deterministic strategy*, which is critical to the computational efficiency.

**2. Improved algorithm for offline two-player zero-sum Markov games.** For offline two-player zero-sum Markov games, we utilize its special structure to develop a maximin-optimization-type algorithm. Though the nonlinear strategy-wise bonus breaks the bilinear structure of the zero-sum game, we show that by solving a maximin optimization problem we can still output a good strategy. In addition, we can solve it efficiently using any black-box algorithms for Lipschitz-continuous convex optimization. Our sample complexity improves the $AB$ factor in Cui and Du [2022] to $(A + B)$.

**3. The first algorithm for offline multi-player general-sum Markov games.** For multi-player general-sum Markov games, we develop a *surrogate function* to approximate performance gap and then show that the minimizer of the surrogate function approximates NE well. The surrogate function is constructed by optimistic best response values and pessimistic values. Interestingly, to our knowledge, this is the first time that optimism has been used in offline RL algorithms. Our result validates that unilateral coverage is sufficient for general-sum Markov games and our sample complexity rate scales with $\widetilde{O}(\sum_{j=1}^{m} A_j)$ (ignoring other parameters), thus breaking the curse of multiagents.

**4. Incorporating pre-specified strategy class.** Lastly, our algorithm allows exploiting the prior knowledge about the NE strategy with an adaptive sample complexity bound. Pre-specified policy class has been widely used in empirical works where the policy class is parameterized by neural networks (e.g., Mnih et al. [2016], Haarnoja et al. [2018], Lowe et al. [2017]), and single-agent RL theory as well (e.g., Auer et al. [2002], Agarwal et al. [2021]), but has not been investigated in MARL theory. In this paper, we take a step to incorporate prior knowledge in the MARL setting. Our performance guarantee only depends on the logarithmic covering number of the pre-specified strategy class, which is always upper bounded by $\sum_{j \in [m]} A_j$, but can be smaller. To the best of our knowledge, this is the first paper that considers a pre-specified strategy class in MARL theory.

## 1.2 Technical Overview of Strategy-wise Concentration

To give some intuition about this technique, let us consider a toy problem. Suppose there are $m$ random variables $\{x^i\}_{i=1}^{m}$ and we want to obtain a pessimistic estimate of their average $x = \sum_{i \in [m]} x^i / m$. We have $n/m$ observations for each $x^i$. The point-wise concentration estimate corresponds to

estimating each $x^i$ and then aggregating the results. The pessimistic estimate of $x^i$ would be $\widehat{x}^i - \widetilde{O}(\sqrt{m/n})$ where $\widehat{x}^i$ is the empirical mean, and the aggregated mean of these pessimistic estimates would be $\widehat{x} - \widetilde{O}(\sqrt{m/n})$ where $\widehat{x}$ is the empirical mean of all data. The strategy-wise concentration estimate corresponds to directly using all the samples to estimate the average of $\{x\}_{i=1}^m$ and obtain the pessimistic estimate as $\widehat{x} - \widetilde{O}(1/\sqrt{n})$. This example shows that the point-wise estimate will lead to an extra $m$ factor. In MARL, $m$ is the cardinality of the joint action space, which implies that point-wise concentration can be exponentially worse than strategy-wise concentration. Note that this is not an issue in single-agent MDP as the optimal policy is always deterministic but leads to severe suboptimality in the multi-agent case where NE can be a mixed strategy.

### 1.3 Related Work

**Online Multi-agent RL.** Markov games can be solved via dynamic programming when the rewards and transition dynamics are given [Hansen et al., 2013, Perolat et al., 2015]. If the environment is unknown, reinforcement learning algorithms are applied with different sampling oracles. One particular line of research is online Markov games, including two-player zero-sum Markov games [Liu et al., 2021, Dou et al., 2021, Xie et al., 2020, Bai et al., 2020, Huang et al., 2021] and multi-player general-sum Markov games [Zhong et al., 2021, Mao et al., 2021, Jin et al., 2021a, Song et al., 2021]. Rubinstein [2016] proves an exponential (in the number of players) lower bound for learning the NE strategy in $m$-player general-sum game while others show that the correlated equilibrium and coarse correlated equilibrium admit $\text{poly}\big(m, \max_{j\in[m]} A_j, H, S\big)$-sample complexity algorithms [Mao et al., 2021, Jin et al., 2021a, Song et al., 2021]. Our upper bounds for $m$-player general-sum games depend polynomially on all parameters, which do not contradict the hardness result in Rubinstein [2016] because the assumptions on the offline dataset provide additional information about the NE.

**Offline Single-agent RL.** The simplest dataset assumption for offline RL is uniform coverage, i.e., the dataset covers all the state-action pairs. This assumption dates back to Szepesvári and Munos [2005]. The minimax sample complexity has been well studied for both tabular case and function approximation [Xie and Jiang, 2021, Yin et al., 2020, 2021, Ren et al., 2021]. Recently it has been shown that only covering the optimal policy is sufficient for offline RL under different settings [Rashidinejad et al., 2021, Yin and Wang, 2021, Xie et al., 2021b, Jin et al., 2021b, Uehara and Sun, 2021, Zanette et al., 2021, Xie et al., 2021a]. These works design provably efficient algorithms based on the principle of pessimism.

**Offline Multi-agent RL.** Offline MARL theory is still at a primary stage. Previous works mostly focused on uniform coverage assumption, i.e. all state-action pairs or all policies are covered [Sidford et al., 2020, Cui and Yang, 2021, Zhang et al., 2020, 2021b, Abe and Kaneko, 2020, Subramanian et al., 2021]. Recently, Cui and Du [2022] and Zhong et al. [2022] show that the unilateral coverage assumption is the minimal dataset coverage assumption for learning NE in Markov games. In addition, [Cui and Du, 2022] proposes a pessimism-type algorithm with $\widetilde{O}(SABH^3C(\pi^*)/\epsilon^2)$ sample complexity for tabular two-player zero-sum Markov game and [Zhong et al., 2022] provides a similar algorithm for linear two-player zero-sum Markov games.

## 2 Preliminaries

**Notations.** We use $D(\mathcal{X})$ to denote the single point distributions over the finite set $\mathcal{X}$. For example, $D(\mathcal{A})$ to represent the policies that deterministically choose one of the actions in $\mathcal{A}$. We use $\pi_{j,h}^s \in \Delta(\mathcal{A}_j)$ as a concise notation of $\pi_{j,h}(\cdot|s)$ and $P_h(s, \mathbf{a})$ to denote $P_h(\cdot|s, \mathbf{a})$, which will be defined in the following section. We use $-j$ in subscript to denote all the players except player $j$. We use bold letter to denote vectors, e.g. $\mathbf{a}$ is a vector and $a_j$ is the $j$-th element of $\mathbf{a}$. We let $O(\cdot)$ hide absolute constants and $\widetilde{O}(\cdot)$ hide polylog terms as well. The L1 norm of a vector in $\mathbb{R}^d$ is $\|\mathbf{a}\|_1 = \sum_{i=1}^d |a_i|$. We denote the projection as $\text{proj}_{[a,b]}(x) := \max\{a, \min\{b, x\}\}$.

**Multi-player General-sum Markov Game.** A multi-player general-sum Markov game is described by a tuple $\mathcal{G} = (\mathcal{S}, \mathcal{A} = \prod_{j\in[m]} \mathcal{A}_j, P, R, H)$, where $\mathcal{S}$ is the state space with cardinality $S$, $m$ is the number of players, $\mathcal{A}_j$ is the action space of player $j$ with cardinality $A_j$, $P = (P_1, P_2, \cdots, P_H)$ with $P_h \in \mathbb{R}^{S \times \prod_{i\in[m]} A_i \times S}$ being the (unknown) transition matrix at timestep $h \in [H]$, $R = \{R_h(\cdot|s_h, \mathbf{a}_h)\}_{h=1}^H$ with $R_h(\cdot|s_h, \mathbf{a}_h)$ being a distribution on $[0,1]^m$ with mean $\mathbf{r}_h(s_h, \mathbf{a}_h) \in [0,1]^m$

as the (unknown) reward distribution at timestep $h$. At timestep $h$, all players choose their actions *simultaneously* and a reward vector is sampled from the reward distribution $\mathbf{r}_h \sim R_h(\cdot|s_h, \mathbf{a}_h)$, where $s_h$ is the current state and $\mathbf{a}_h = (a_{h,1}, a_{h,2}, \cdots, a_{h,m})$ is the joint action. Each player $j$ receives its own reward $r_{h,j}$ with support on $[0,1]$ and mean $r_{h,j}(s_h, \mathbf{a}_h)$. The state then transits to $s_{h+1}$ following the distribution of $P_h(\cdot \mid s_h, \mathbf{a}_h)$. The game terminates at timestep $H+1$. We assume that the initial state $s_1$ is fixed because for a stochastic initial state, one can add $s_0$ as the initial state instead and it transits to $s_1$ following the initial distribution.

We denote a joint strategy as $\pi = (\pi_1, \pi_2, \cdots, \pi_m)$, where $\pi_j = (\pi_{1,j}, \pi_{2,j}, \cdots, \pi_{H,j})$ and $\pi_{h,j} : \mathcal{S} \to \Delta(\mathcal{A}_j)$ is the strategy of player $j$ at timestep $h$ where $\Delta(\mathcal{A}_j)$ is the probability simplex over $\mathcal{A}_j$. We use $\Pi^{\text{full}}$ to denote the set of all the possible joint strategies. We define the state value function and state-action value function under strategy $\pi$ for each player $j \in [m]$:

$$V_{h,j}^{\pi}(s_h) := \mathbb{E}_{\pi}\left[ \sum_{t=h}^{H} r_{t,j}(s_t, \mathbf{a}_t) \,\middle|\, s_h \right], Q_{h,j}^{\pi}(s_h, \mathbf{a}_h) := \mathbb{E}_{\pi}\left[ \sum_{t=h}^{H} r_{t,j}(s_t, \mathbf{a}_t) \,\middle|\, s_h, \mathbf{a}_h \right],$$

where the expectation is over the randomness of the environment and the joint strategy $\pi$. For a fixed player $j$, if all the other player's strategies are fixed, then player $j$ can play the best response strategy to maximize its own total reward. We define $\pi_{-j}$ to be the strategy for all players except player $j$ and define the best response value to be $V_{h,j}^{*,\pi_{-j}}(s_h) := \max_{\pi_j} V_{h,j}^{\pi_j, \pi_{-j}}(s_h)$.

It is well-known that Nash equilibrium strategy exists for general-sum Markov games. Note that there could be multiple NE strategies with different value functions. We use the following performance gap to evaluate a strategy $\pi$: $\text{Gap}(\pi) := \sum_{j \in [m]}\left[ V_{1,j}^{*,\pi_{-j}}(s_1) - V_{1,j}^{\pi}(s_1) \right]$. This metric is always non-negative and we say $\pi$ is an $\epsilon$-approximate NE if and only if $\text{Gap}(\pi) \le \epsilon$.

**Two-player Zero-sum Markov Game.** A general-sum Markov game becomes a two-player zero-sum Markov game if there are only two players and the reward $r_h \sim R_h(\cdot|s, a_1, a_2)$ always satisfies $r_{h,1} + r_{h,2} = 0$ for all $h \in [H]$, $s \in \mathcal{S}$, $a_1 \in \mathcal{A}_1$ and $a_2 \in \mathcal{A}_2$. Following the literatures on two-player zero-sum Markov games, we use slightly different notations for this setting. There is only one reward function $r$ shared by both players, which is the reward function $\{r_{h,1}\}_{h=1}^{H}$ for player 1 and the target of player 2 is to minimize the total reward. We denote $\mu = \pi_1$ and $\nu = \pi_2$ to be the strategy for each player, $a = a_1$ and $b = a_2$ to be the action for each player, $\Pi^{\max} = \Pi_1$ and $\Pi^{\min} = \Pi_2$ to be the strategy class for each player to remove extra subscripts. One can derive the performance gap under the new notations for two-player zero-sum Markov games: $\text{Gap}(\pi) := V_1^{*,\nu}(s_1) - V_1^{\mu,*}(s_1)$.

**Offline Markov Game.** In offline RL, the dataset is collected beforehand and no further sampling is allowed. Here we consider offline multi-player general-sum Markov game. The framework for offline two-player zero-sum Markov game is similar with the slightly different notations as we mentioned.

We assume that the algorithm has access to an offline dataset $\mathcal{D} = \{(s_h^k, \mathbf{a}_h^k, \mathbf{r}_h^k, s_{h+1}^k)\}_{h,k=1,1}^{H,n}$ that satisfies Assumption 1. The assumption states that the dataset is independently generated from the underlying Markov game, which is used in [Jin et al., 2021b, Zhong et al., 2022]. The target of offline Markov game is to find a strategy $\pi$ with as small performance gap as possible by utilizing the dataset $\mathcal{D}$. One closely related assumption is that the dataset is generated from some behavior strategy [Xie et al., 2021b, Cui and Du, 2022]. Though this kind of dataset does not satisfy Assumption 1 directly due to the dependence within the trajectory, we can construct a compliant dataset by using the subsampling technique in Li et al. [2022] while the number of samples is still of the same order.

**Assumption 1.** *The dataset $\mathcal{D}$ is compliant with the multi-player general-sum markov game, i.e.,*

$$\mathbb{P}_{\mathcal{D}}(s_{h+1}^k = s \mid s_h^k, \mathbf{a}_h^k) = P_h(s_{h+1} = s \mid s_h = s_h^k, \mathbf{a}_h = \mathbf{a}_h^k),$$

$$\mathbb{P}_{\mathcal{D}}(\mathbf{r}_h^k = \mathbf{r} \mid s_h^k, \mathbf{a}_h^k) = R_h(\mathbf{r}_h = \mathbf{r} \mid s_h = s_h^k, a_h = \mathbf{a}_h^k), \forall j \in [m],$$

*for all $h \in [H]$ and $k \in [n]$. In addition, all tuples $(s_h^k, \mathbf{a}_h^k, \mathbf{r}_h^k, s_{h+1}^k)$ are independent.*

**Pre-specified Policy Class.** We also consider the case when we know that the NE is possibly in a given subset of $\Pi^{\text{full}}$. We denote this subset as $\Pi$ and our target is to find the best strategy in $\Pi$. Note that we do not assume NE is indeed in $\Pi$. In addition, by choosing $\Pi = \Pi^{\text{full}}$ we can recover the standard setting. To measure the complexity of $\Pi$, we use the covering number.

**Definition 1.** *(Covering Number) For any error level $\epsilon_{\text{cover}}$ and strategy class $\Pi$, we define*

$$\mathcal{N}(\Pi, \epsilon_{\text{cover}}) := \sum_{s \in \mathcal{S}, h \in [H]} \prod_{j \in [m]} |\mathcal{C}(\Pi_{h,j}(s), \epsilon_{\text{cover}})|,$$

---

**Algorithm 1** **S**trategy-wise **B**onus + **M**axi**M**in Optimization (SBMM)

---

Input: offline dataset $\mathcal{D}$.

Initialization: $\underline{V}_{H+1}(s) = \overline{V}_{H+1}(s) = 0$ for all $s \in \mathcal{S}$.

**for** time $h = H, H-1, \ldots, 1$ **do**

  #Player 1

  Approximately solve $\underline{\mu}_h^s = \arg\max_{\mu_h^s \in \Pi_h^{\max}(s)} \min_{\nu_h^s \in D(\mathcal{B})} \underline{V}_h^{\mu_h^s, \nu_h^s}(s)$, where $\underline{V}_h^{\mu_h^s, \nu_h^s}(s)$ is defined by (4) and (5) and $\underline{\mu}_h^s$ satisfies (10).

  Solve $\underline{\nu}_h^s = \arg\min_{\nu_h^s \in D(\mathcal{B})} \underline{V}_h^{\underline{\mu}_h^s, \nu_h^s}(s)$ and set $\underline{V}_h(s) = \text{proj}_{[0, H-h+1]}\left\{\underline{V}_h^{\underline{\mu}_h^s, \underline{\nu}_h^s}(s)\right\}$.

  #Player 2

  Approximately solve $\overline{\nu}_h^s = \arg\min_{\nu_h^s \in \Pi_h^{\min}(s)} \max_{\mu_h^s \in D(\mathcal{A})} \overline{V}_h^{\mu_h^s, \nu_h^s}(s)$, where $\overline{V}_h^{\mu_h^s, \nu_h^s}(s)$ is defined by (8) and (9) and $\overline{\nu}_h^s$ satisfies (11).

  Solve $\overline{\mu}_h^s = \arg\max_{\mu_h^s \in D(\mathcal{A})} \overline{V}_h^{\mu_h^s, \overline{\nu}_h^s}(s)$ and set $\overline{V}_h^s = \text{proj}_{[0, H-h+1]}\left\{\overline{V}_h^{\overline{\mu}_h^s, \overline{\nu}_h^s}(s)\right\}$.

**end for**

Output $\pi^{\text{output}} = (\underline{\mu}, \overline{\nu})$.

---

where $\Pi_{h,j}(s) = \{\pi_h^j(\cdot|s) : \pi \in \Pi\}$ is a subset of $\Delta(\mathcal{A}_i)$ and $\mathcal{C}(\Pi_{h,j}(s), \epsilon_{\text{cover}})$ is an $\epsilon_{\text{cover}}$-covering of $\Pi_{h,j}(s)$ with respect to the L1 norm $\|\cdot\|_1$.

Our performance guarantee will only have logarithm dependence on $\mathcal{N}(\Pi, \epsilon_{\text{cover}})$. As $\Pi_{h,j}(s)$ is a subset of $\Delta(\mathcal{A}_j)$, we always have $\log(\mathcal{N}(\Pi, \epsilon_{\text{cover}})) \leq \widetilde{O}(\sum_{j \in [m]} A_j \log(1/\epsilon_{\text{cover}}))$ and if $\Pi$ is a finite set, we have $\log(\mathcal{N}(\Pi, \epsilon_{\text{cover}})) \leq \log(SH|\Pi|)$ (see Appendix B.1 for the proof). In this paper we will choose $\epsilon_{\text{cover}} = \frac{1}{\sum_{j \in [m]} A_j m H^2 n^2}$, which only leads to logarithm dependence on these quantities. In later sections, we will omit $\epsilon_{\text{cover}}$ to simplify the notation.

For any joint strategy $\pi$, we call $(\pi_j', \pi_{-j})$ for any strategy $\pi'$ and $j \in [m]$ as a unilateral strategy of $\pi$. Previous works show that only covering an NE is not sufficient, and covering all the unilateral strategies of an NE is necessary for learning the NE in Markov games [Cui and Du, 2022, Zhong et al., 2022]. We use unilateral coefficient to quantify how the dataset covers all the unilateral strategies of a strategy $\pi$. If we assume that the dataset is sampled from some (unknown) distribution, i.e. $(s_h, \mathbf{a}_h) \sim d_h(\cdot, \cdot)$ for all $h \in [H]$, we can define the population unilateral coefficient.

**Definition 2.** *For any strategy $\pi$, the population unilateral coefficient is defined as* $C(\pi) := \max_{h, j, \pi', s_h, \mathbf{a}_h} \frac{d_h^{\pi_j', \pi_{-j}}(s_h, \mathbf{a}_h)}{d_h(s_h, \mathbf{a}_h)}$.

Cui and Du [2022] provide a sample complexity result for zero-sum Markov games with dependence on $C(\pi^*)$. We can also define the empirical unilateral coefficient using the empirical distribution.

**Definition 3.** *Define the empirical dataset distribution as* $\widehat{d}_h(s, \mathbf{a}) = n_h(s, \mathbf{a})/n$, *for all* $h \in [H], s \in \mathcal{S}, \mathbf{a} \in \mathcal{A}$, *where* $n_h(s, \mathbf{a})$ *is the number of times that* $(s, \mathbf{a})$ *appears in the dataset for timestep* $h$. *For any strategy $\pi$, the empirical unilateral coefficient is defined as* $\widehat{C}(\pi) := \max_{h, j, \pi', s_h, \mathbf{a}_h} \frac{d_h^{\pi_j', \pi_{-j}}(s_h, \mathbf{a}_h)}{\widehat{d}_h(s_h, \mathbf{a}_h)}$.

The empirical unilateral coefficient can lead to dataset-dependent bound that has no dependence on the underlying distribution of the dataset. In addition, $\widehat{C}(\pi)$ can be bounded by $2C(\pi)$ (Proposition 1) so results based on $\widehat{C}(\pi)$ directly transfer to $C(\pi)$. Note that $\widehat{C}(\pi)$ and $C(\pi)$ are both unknown to the algorithm and only appear in the analysis and theorems.

**Proposition 1.** *Suppose* $p_{\min} = \min_{s, \mathbf{a}, h}\{d_h(s, \mathbf{a}) : d_h(s, \mathbf{a}) > 0\}$. *If* $n \geq \frac{8 \log(S \Pi_{j \in [m]} A_j H/\delta)}{p_{\min}}$, *with probability* $1 - \delta$, *for all strategy $\pi$, we have* $2C(\pi) \geq \widehat{C}(\pi)$.

## 3  An Improved Algorithm for Offline Two-player Zero-sum Markov Game

In this section, we propose a new algorithm for offline zero-sum Markov game based on two novel techniques, i.e., strategy-wise concentration and maximin-optimization-based algorithm. We then show that this algorithm is computationally efficient and can (almost) find the best strategy in strategy class $\Pi$ with favorable sample complexity.

Let us first define some notations. Given a dataset $\mathcal{D} = \{(s_h^k, a_h^k, b_h^k, r_h^k, s_{h+1}^k)\}_{k,h=1}^{n,H}$, we denote $n_h(s,a,b) = \sum_{k=1}^{n} \mathbf{1}\left((s_h^k, a_h^k, b_h^k) = (s,a,b)\right)$ and $\mathcal{K}_h(s) = \{(a,b) \in \mathcal{A} \times \mathcal{B} : n_h(s,a,b) \neq 0\}$. If $n_h(s,a,b) \neq 0$, we set

$$\widehat{r}_h(s,a,b) = \frac{\sum_{k=1}^{n} r_h^k \mathbf{1}\left((s_h^k, a_h^k, b_h^k) = (s,a,b)\right)}{n_h(s,a,b)}, \tag{1}$$

$$\widehat{P}_h(s'|s,a,b) = \frac{\sum_{k=1}^{n} \mathbf{1}\left((s_h^k, a_h^k, b_h^k, s_{h+1}^k) = (s,a,b,s')\right)}{n_h(s,a,b)}, \tag{2}$$

otherwise we have

$$\widehat{r}_h(s,a,b) = 0, \widehat{P}_h(s'|s,a,b) = 0. \tag{3}$$

Based on this empirical Markov game, we can perform value-iteration-type algorithm. Here we describe our algorithm for player 1. For each timestep $h$, we first compute the the state-action values based on the estimates at timestep $h+1$:

$$\underline{Q}_h(s,a,b) = \widehat{r}_h(s,a,b) + \left\langle \widehat{P}_h(s,a,b), \underline{V}_{h+1} \right\rangle, \tag{4}$$

Then instead of adding the bonus on state-action estimates directly to ensure pessimism as used in Cui and Du [2022] and Zhong et al. [2022], we first estimate the state value functions for strategy $\mu_h^s, \nu_h^s$ and then add the bonus on them instead.

$$\underline{V}_h^{\mu_h^s, \nu_h^s}(s) = \mathbb{E}_{a \sim \mu_h^s, b \sim \nu_h^s} \underline{Q}_h(s,a,b) - b_h(s, \mu_h^s, \nu_h^s), \tag{5}$$

where

$$b_h(s, \mu_h^s, \nu_h^s) = H \sqrt{\sum_{(a,b) \in \mathcal{K}_h(s)} \frac{\mu_h^s(a)^2 \nu_h^s(b)^2}{n_h(s,a,b)} \log(\mathcal{N}(\Pi))\iota} + \sqrt{\iota}/n, \tag{6}$$

with $\iota = 32 \log(2ABSHn/\delta)$. We also present the bonus from point-wise concentration used in Cui and Du [2022] to better compare them, $b_h^{\text{point}}(s, \mu_h^s, \nu_h^s) = H \sum_{(a,b) \in \mathcal{K}_h(s)} \mu_h^s(a) \nu_h^s(b) \sqrt{\frac{\iota}{n_h(s,a,b)}}$.

As a concrete example, if $\mu_h^s$ and $\nu_h^s$ are uniform distribution on $\mathcal{A}$ and $\mathcal{B}$, then $b_h(s, \mu_h^s, \nu_h^s)$ is smaller than $b_h^{\text{point}}(s, \mu_h^s, \nu_h^s)$ for an order of $\sqrt{AB}$. Finally to obtain the pessimistic value estimate, we solve the following optimization problem

$$\underline{V}_h(s) = \max_{\mu_h^s \in \Pi_h^{\max}(s)} \min_{\nu_h^s \in D(\mathcal{B})} \underline{V}_h^{\mu_h^s, \nu_h^s}(s). \tag{7}$$

Here recall that $D(\mathcal{B})$ represents all the deterministic strategies in $\mathcal{B}$. Our algorithm is similar for player 2 with the following $\overline{Q}$ and $\overline{V}$ estimation:

$$\overline{Q}_h(s,a,b) = \widehat{r}_h(s,a,b) + \left\langle \widehat{P}_h(s,a,b), \overline{V}_{h+1} \right\rangle + H\mathbf{1}\{(a,b) \notin \mathcal{K}_h(s)\}, \tag{8}$$

$$\overline{V}_h^{\mu_h^s, \nu_h^s}(s) = \mathbb{E}_{\mu_h^s, \nu_h^s} \overline{Q}_h(s,a,b) + b_h(s, \mu_h^s, \nu_h^s). \tag{9}$$

The additional $H\mathbf{1}\{(a,b) \notin \mathcal{K}_h(s)\}$ term in (8) compared with (4) is to compensate the underestimate by (3).

## 3.1 Computational Efficiency

For computational efficiency, we start with the following characterization about our bonus.

**Proposition 2.** $\underline{V}_h^{\mu_h^s, \nu_h^s}(s)$ is concave and $\overline{V}_h^{\mu_h^s, \nu_h^s}(s)$ is convex w.r.t. $\mu_h^s$ and $\nu_h^s$ respectively.

Proposition 2 explains why the inner minimization in (7) is over the deterministic strategy class as the minimum of a concave function over the probability simplex is achieved at the vertexes, i.e. deterministic strategies. The proof of Proposition 2 is provided in Appendix B.2.

Previous works solve the NE (saddle point) of $\underline{V}_h^{\mu_h^s, \nu_h^s}(s)$ as the point-wise bonus maintains the bilinear structure [Cui and Du, 2022, Zhong et al., 2022]. Though here $\underline{V}_h^{\mu_h^s, \nu_h^s}(s)$ no longer enjoys

the strong duality, we will show that solving the maximin problem is enough to obtain a good strategy for player 1. As the inner minimization is only on a feasible set of size $B$, this problem can be solved efficiently by using projected gradient descent [Bubeck et al., 2015]. We assume that we solve the maximin and the minimax optimization problem to $\epsilon_{\text{opt}}$-optimality, i.e.

$$\min_{\nu_h^s \in D(\mathcal{B})} \underline{V}_h^{\mu_h^s, \nu_h^s}(s) \geq \max_{\mu_h^s \in \Pi_h^{\max}(s)} \min_{\nu_h^s \in D(\mathcal{B})} \underline{V}_h^{\mu_h^s, \nu_h^s}(s) - \epsilon_{\text{opt}}, \tag{10}$$

$$\max_{\mu_h^s \in D(\mathcal{A})} \overline{V}_h^{\mu_h^s, \overline{\nu}_h^s}(s) \leq \min_{\nu_h^s \in \Pi_h^{\min}(s)} \max_{\mu_h^s \in D(\mathcal{A})} \overline{V}_h^{\mu_h^s, \nu_h^s}(s) + \epsilon_{\text{opt}}. \tag{11}$$

In Appendix B.2 we show that projected gradient descent can output an $\epsilon_{\text{opt}}$-minimizer with $(H + H\sqrt{\log(\mathcal{N}(\Pi))\iota})/\epsilon_{\text{opt}}^2$ iterations, where each iteration consists of a gradient computation and a projection onto the probability simplex. We note that if we set $\epsilon_{\text{opt}}$ to $\frac{1}{\sqrt{n}}$, then the optimization error is always of a smaller order term compared to the statistical error.

## 3.2 Sample Complexity Guarantees for **SBMM**

For the statistical guarantee, we will first provide *assumption-free bounds* in the sense that it holds for arbitrary compliant dataset [Jin et al., 2021b, Yin and Wang, 2021]. We define the uncertainty at timestep $h$ and state $s$ under strategy $\mu_h^s$ and $\nu_h^s$:

$$\widehat{b}_h(s, \mu_h^s, \nu_h^s) := 2b_h(s, \mu_h^s, \nu_h^s) + H \sum_{(a,b) \notin \mathcal{K}_h(s)} \mu_h^s(a)\nu_h^s(b)$$

.

**Proposition 3.** *Suppose $\pi^{\text{output}}$ is the output of Algorithm 1. With probability $1 - \delta$, we have* $\text{Gap}(\pi^{\text{output}}) \leq$

$$\min_{\pi = (\mu, \nu) \in \Pi} \max_{\pi' = (\mu', \nu') \in \Pi^{\text{det}}} \left[ \text{Gap}(\pi) + \mathbb{E}_{\mu, \nu'} \sum_{h=1}^H \widehat{b}_h(s_h, \mu_h^{s_h}, \nu_h^{'s_h}) + \mathbb{E}_{\mu', \nu} \sum_{h=1}^H \widehat{b}_h(s_h, \mu_h^{'s_h}, \nu_h) \right] + 2H\epsilon_{\text{opt}}.$$

Proposition 3 shows that our algorithm can find the best strategy in $\Pi$ with an additional error of the expected total uncertainty under some unilateral strategies and an extra optimization error term $2H\epsilon_{\text{opt}}$. Then we derive bounds with unilateral coefficients.

**Theorem 1.** *Suppose $\pi^{\text{output}}$ is the output of Algorithm 1. With probability $1 - \delta$, we have*

$$\text{Gap}(\pi^{\text{output}}) \leq \min_{\pi \in \Pi} \left[ \text{Gap}(\pi) + 4H^2 \sqrt{S \log(\mathcal{N}(\Pi))\widehat{C}(\pi)\iota/n} \right] + 2H\epsilon_{\text{opt}}.$$

Theorem 1 directly implies the following corollary.

**Corollary 1.** *If $\Pi = \Pi^{\text{full}}$, then with probability $1 - \delta$, we have* $\text{Gap}(\pi^{\text{output}}) = \widetilde{O}(\sqrt{H^4 S(A + B)\widehat{C}(\pi^*)/n}) + 2H\epsilon_{\text{opt}}$. *If $\pi^* \in \Pi$, then with probability $1 - \delta$, we have* $\text{Gap}(\pi^{\text{output}}) = \widetilde{O}(\sqrt{H^4 S \log(\mathcal{N}(\Pi))\widehat{C}(\pi^*)/n}) + 2H\epsilon_{\text{opt}}$.

Since $\widehat{C}(\pi)$ can be bounded using $C(\pi)$ (Proposition 1), we have the following theorem.

**Theorem 2.** *Suppose $\pi^{\text{output}}$ is the output of Algorithm 1. With probability $1 - \delta$, we have*

$$\text{Gap}(\pi^{\text{output}}) \leq \min_{\pi \in \Pi} \left[ \text{Gap}(\pi) + 4H^2 \sqrt{S \log(\mathcal{N}(\Pi))C(\pi)\iota^2/n} + HS(A + B)C(\pi)/n \right] + 2H\epsilon_{\text{opt}}.$$

*In addition, suppose $p_{\min} = \min_{s,a,b,h}\{d_h^\rho(s, a, b) : d_h^\rho(s, a, b) > 0\}$ and if $n \geq \frac{8\log(SABH/\delta)}{p_{\min}}$, we have* $\text{Gap}(\pi) \leq \min_{\pi \in \Pi} \left[ \text{Gap}(\pi) + 8H^2 \sqrt{S \log(\mathcal{N}(\Pi))C(\pi)\iota^2/n} \right] + 2H\epsilon_{\text{opt}}.$

Theorem 2 shows that there will be an additional lower order term $S(A + B)C(\pi)/n$, which can be interpreted as the rate of the empirical dataset distribution converges to the population distribution. In addition, for large enough $n \geq \frac{8\log(SABH/\delta)}{p_{\min}}$, there is no lower order term. Here $n \geq \frac{8\log(SABH/\delta)}{p_{\min}}$ serves as a warm-up cost so that the empirical support is the same as the true support of $d_h$. A similar analysis is used in Yin and Wang [2021]. With a refined analysis, we can show that there is no lower order term for the standard settings $\Pi = \Pi^{\text{full}}$ in two-player zero-sum Markov games and $\Pi = \Pi^{\text{det}}$ for turn-based Markov games. Note that turn-based Markov games always have a deterministic NE.

**Corollary 2.** *If $\Pi = \Pi^{\text{full}}$, then with probability $1 - \delta$, we have* $\text{Gap}(\pi^{\text{output}}) = \widetilde{O}(\sqrt{H^4 S(A + B)C(\pi^*)/n}) + 2H\epsilon_{\text{opt}}$. *In addition, for turn-based two-player zero-sum Markov games, we can set $\Pi = \Pi^{\text{det}}$ and we have* $\text{Gap}(\pi^{\text{output}}) = \widetilde{O}(\sqrt{H^4 S C(\pi^*)/n}) + 2H\epsilon_{\text{opt}}$.

Corollary 2 improves the $AB$ dependence in the previous zero-sum Markov games result [Cui and Du, 2022] and matches the result for turn-based Markov games [Cui and Du, 2022] up to an extra $\sqrt{H}$ factor. The additional $H$ factor is due to the Hoeffding-type bonus and we believe it can be removed with a more sophisticated Bernstein-type bonus.

## 4 Algorithms and Analyses for Multi-player General-sum Markov Game

In this section, we propose the first provably efficient algorithm for offline multi-player general-sum Markov game. We will use the strategy-wise bonus to achieve a sample complexity that does not scale with $\prod_{j\in[m]} A_j$. However, in general-sum games there is no saddle point structure, so we can no longer use the maximin-optimization-type algorithm. Instead, our algorithm utilizes a novel *surrogate function* to approximately minimize the performance gap.

Given a dataset $\mathcal{D} = \{(s_h^k, \mathbf{a}_h^k, \mathbf{r}_h^k, s_{h+1}^k)\}_{k,h=1}^{n,H}$, we denote $n_h(s, \mathbf{a}) = \sum_{k=1}^n \mathbf{1}\left((s_h^k, \mathbf{a}_h^k) = (s, \mathbf{a})\right)$ and $\mathcal{K}_h(s) = \{\mathbf{a} : n_h(s, \mathbf{a}) \neq 0\}$. If $n_h(s, \mathbf{a}) > 0$, we set

$$\widehat{r}_{h,j}(s, \mathbf{a}) = \frac{\sum_{k=1}^n r_{h,j}^k \mathbf{1}\left((s_h^k, \mathbf{a}_h^k) = (s, \mathbf{a})\right)}{n_h(s, \mathbf{a})}, \widehat{P}_h(s'|s, \mathbf{a}) = \frac{\sum_{k=1}^n \mathbf{1}\left((s_h^k, \mathbf{a}_h^k, s_{h+1}^k) = (s, \mathbf{a}, s')\right)}{n_h(s, \mathbf{a})}, \tag{12}$$

otherwise we have $\widehat{r}_{h,j}(s, \mathbf{a}) = 0, \widehat{P}_h(s'|s, \mathbf{a}) = 0$.

Based on this empirical multi-player Markov game, we can estimate the value of arbitrary strategy $\pi$ via policy evaluation (Algorithm 2 in Appendix). We describe Algorithm 2 for the pessimistic estimate. For a player $j$, strategy $\pi$ and timestep $h$, we first compute the state-action value estimates:

$$\underline{Q}_{h,j}^\pi(s, \mathbf{a}) = \widehat{r}_{h,j}(s, \mathbf{a}) + \left\langle \widehat{P}_h(s, \mathbf{a}), \underline{V}_{h+1,j}^\pi \right\rangle, \tag{13}$$

Then we estimate the state value functions and add the strategy-wise bonus to ensure pessimism.

$$\underline{V}_{h,j}^\pi(s) = \text{proj}_{[0,H-h+1]}\left\{ \mathbb{E}_{\mathbf{a}\sim\pi_h(\cdot|s)}\underline{Q}_{h,j}^\pi(s, \mathbf{a}) - b_h(s, \pi_h^s) \right\}, \tag{14}$$

$$\text{where } b_h(s, \pi_h^s) = H\sqrt{\sum_{\mathbf{a}\in\mathcal{K}_h(s)} \frac{\prod_{j\in[m]} \pi_{h,j}^s(a_j)^2}{n_h(s, \mathbf{a})} S\log(\mathcal{N}(\Pi))\iota} + \sqrt{\iota}/n, \tag{15}$$

with $\iota = 32\log(16\prod_{j\in[m]} A_j mSHn/\delta)$. Here the strategy-wise pessimism can remove the $\prod_{j\in[m]} A_j$ dependence as explained in the previous section. By dynamic programming from timestep $H$ to timestep 1 we can obtain the pessimistic estimate $\underline{V}_{1,j}^\pi(s_1)$. Compared with the bonus function (6) in zero-sum Markov game, there is an extra $S$ factor in (15) because here we need to perform concentration on $\left\langle \widehat{P}_h(s, \mathbf{a}), \underline{V}_{h+1,j}^\pi \right\rangle$ for all $\pi$ while in (4) we only need to analyze $\left\langle \widehat{P}_h(s, a, b), \underline{V}_{h+1} \right\rangle$ for a single $\underline{V}_{h+1}$. We use an additional $\epsilon$-covering on $\mathbb{R}^S$ which leads to the extra $S$.

We use Algorithm 3 (in Appendix) to compute the optimistic value of the best response strategy. For a given player $j$, strategy $\pi_{-j}$ used by all the other player and timestep $h$, we first compute the optimistic state-action value estimate:

$$\overline{Q}_{h,j}^{*,\pi_{-j}}(s, \mathbf{a}) = \widehat{r}_{h,j}(s, \mathbf{a}) + \left\langle \widehat{P}_h(s, \mathbf{a}), \overline{V}_{h+1,j}^{*,\pi_{-j}} \right\rangle + H\mathbf{1}\{\mathbf{a} \notin \mathcal{K}_h(s)\}. \tag{16}$$

Then we compute the optimistic value for deterministic strategies for player $j$:

$$\overline{V}_{h,j}(s, a_j) = \mathbb{E}_{\mathbf{a}_{-j}\sim\pi_{h,-j}(\cdot|s)}\overline{Q}_{h,j}^{*,\pi_{-j}}(s, a_j, \mathbf{a}_{-j}) + b_h(s, a_j, \pi_{h,-j}^s). \tag{17}$$

Here with a slight abuse of the notation, we use $a_j$ to denote the deterministic strategy of player $j$ that chooses action $a_j$ at state $s$ and timestep $h$. Finally we use the maximum over all the deterministic strategies to be the best response value function: $\overline{V}_{h,j}^{*,\pi_{-j}}(s) = \text{proj}_{[0,H-h+1]}\left\{\max_{a_j\in\mathcal{A}_j} \overline{V}_{h,j}(s, a_j)\right\}$.

By dynamic programming we can obtain the optimistic estimate $\overline{V}_{1,j}^{*,\pi_{-j}}(s_1)$ at the initial state. Note that we only consider the deterministic strategies for player $j$. Thanks to the convexity of the bonus $b_h(s,\pi_h^s)$, the best response with respect to $\overline{V}_{h,j}^\pi(s)$ is also in the deterministic strategy class as in zero-sum Markov games. The following proposition connects Algorithm 2 and Algorithm 3:

**Proposition 4.** *For any strategy* $\pi_{-j} \in \Pi_{-j}^{\mathrm{full}}, h \in [H]$ *and* $s \in \mathcal{S}$, *we have* $\overline{V}_{h,j}^{*,\pi_{-j}}(s) = \max_{\pi_j} \overline{V}_{h,j}^{\pi_j,\pi_{-j}}(s)$.

Based on Algorithm 2 and Algorithm 3, we propose a surrogate minimization algorithm for multi-player general-sum Markov game. Suppose $\underline{V}_{1,j}^\pi(s_1)$ and $\overline{V}_{1,j}^{*,\pi_{-j}}(s_1)$ are pessimistic and optimistic estimates, then we have

$$\mathrm{Gap}(\pi) = \sum_{j \in [m]} V_{1,j}^{*,\pi_{-j}}(s_1) - V_{1,j}^\pi(s_1) \leq \sum_{j \in [m]} \overline{V}_{1,j}^{*,\pi_{-j}}(s_1) - \underline{V}_{1,j}^\pi(s_1).$$

The RHS can serve as the surrogate function and SBSM (Algorithm 4 in Appendix) outputs the minimizer of it in $\Pi$. From the computational perspective, Algorithm 2 and Algorithm 3 are both efficient while Algorithm 4 needs to enumerate $\Pi$ for the worst case. This computational hardness agrees with the PPAD-hardness for computing approximate NE even in full information general-sum game [Daskalakis, 2013]. However, if $\Pi$ is well structured, Algorithm 4 may be computationally efficient and we leave it to future work. Here we assume $\pi^{\mathrm{output}}$ is an exact solution while it is straightforward to incorporate optimization error as in the previous section.

## 4.1 Sample Complexity Guarantees for SBSM

We still begin with assumption-free bound as in the previous section. We define the uncertainty at timestep $h$ and state $s$ under strategy $\pi$: $\widehat{b}_h(s,\pi_h^s) = 2b_h(s,\pi_h^s) + H\sum_{\mathbf{a}\notin\mathcal{K}_h(s)}\pi_h^s(\mathbf{a})$.

**Proposition 5.** *Suppose* $\pi^{\mathrm{output}}$ *is the output of Algorithm 4. With probability* $1-\delta$, *we have*

$$\mathrm{Gap}(\pi^{\mathrm{output}}) \leq \min_{\pi \in \Pi}\left[\mathrm{Gap}(\pi) + \max_{\pi' \in \Pi^{\det}}\sum_{j\in[m]}\mathbb{E}_{\pi_j',\pi_{-j}^*}\sum_{h=1}^H \widehat{b}_h(s_h,\pi_{h,j}'^{s_h},\pi_{h,-j}^{s_h}) + m\mathbb{E}_\pi\sum_{h=1}^H \widehat{b}_h(s_h,\pi_h^{s_h})\right].$$

Proposition 5 has a similar structure as Proposition 3 with a slight difference in the expected uncertainty terms. Then we will bound using the unilateral coefficients.

**Theorem 3.** *Suppose* $\pi^{\mathrm{output}}$ *is the output of Algorithm 4. With probability* $1-\delta$, *we have*

$$\mathrm{Gap}(\pi^{\mathrm{output}}) \leq \min_{\pi \in \Pi}\left[\mathrm{Gap}(\pi) + 4mH^2S\sqrt{\widehat{C}(\pi)\log(\mathcal{N}(\Pi))\iota/n}\right].$$

Theorem 3 directly implies the following corollary, which shows that the sample complexity of offline multi-agent RL only scales linearly with respect to the number of the players.

**Corollary 3.** *If* $\Pi = \Pi^{\mathrm{full}}$, *with probability* $1-\delta$, *we have* $\mathrm{Gap}(\pi^{\mathrm{output}}) = \widetilde{O}(\sqrt{H^4S^2\sum_{j\in[m]}A_j\widehat{C}(\pi^*)/n})$. *If* $\pi^* \in \Pi$, *then with probability* $1-\delta$, *we have* $\mathrm{Gap}(\pi^{\mathrm{output}}) = \widetilde{O}(\sqrt{H^4S^2\log(\mathcal{N}(\Pi))\widehat{C}(\pi^*)/n})$.

Similarly we have the following theorem and corollary for the population unilateral coefficient.

**Theorem 4.** *Suppose* $\pi^{\mathrm{output}}$ *is the output of Algorithm 4. If* $n \geq \frac{8\log(S\Pi_{j\in[m]}A_jH/\delta)}{p_{\min}}$, *with probability* $1-\delta$, *we have* $\mathrm{Gap}(\pi^{\mathrm{output}}) \leq \min_{\pi\in\Pi}\left[\mathrm{Gap}(\pi) + 4mH^2S\sqrt{2C(\pi)\log(\mathcal{N}(\Pi))\iota/n}\right]$.

**Corollary 4.** *Suppose* $n \geq \frac{8\log(S\Pi_{j\in[m]}A_jH/\delta)}{p_{\min}}$. *If* $\Pi = \Pi^{\mathrm{full}}$, *with probability* $1-\delta$, *we have* $\mathrm{Gap}(\pi^{\mathrm{output}}) = \widetilde{O}(\sqrt{H^4S^2\sum_{j\in[m]}A_jC(\pi^*)/n})$. *If* $\pi^* \in \Pi$, *then with probability* $1-\delta$, *we have* $\mathrm{Gap}(\pi^{\mathrm{output}}) = \widetilde{O}(\sqrt{H^4S^2\log(\mathcal{N}(\Pi))C(\pi^*)/n})$.

## 5 Conclusion

In this work, we studied offline MARL. With a novel strategy-wise bonus, we remove the exponential dependence on the number of players. We use different algorithm frameworks for zero-sum Markov games and general-sum Markov games due to their different properties.

Here we list several open problems for future work. One direction is to find the minimax sample complexity for offline Markov games, i.e., if the $\log(\mathcal{N}(\Pi))$ term is necessary. Another direction is to design computationally efficient algorithms for finding (coarse) correlated equilibrium in general-sum Markov games. Lastly, we only focus on the tabular setting serving as a start point. It is important to study MARL with reasonable function approximation.

## Acknowledgements

This work was supported in part by NSF CCF 2212261, NSF IIS 2143493, NSF DMS-2134106, NSF CCF 2019844 and NSF IIS 2110170.

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
