# A Algorithms

---

**Algorithm 2** Value Estimation

---

Input: offline dataset $\mathcal{D}$, player index $j$ and strategy $\pi$.

Initialization: $\underline{V}_{H+1,j}^{\pi}(s) = \overline{V}_{H+1,j}^{\pi}(s) = 0$ for all $s \in \mathcal{S}$.

**for** time $h = H, H-1, \ldots, 1$ **do**

    Set $\underline{Q}_{h,j}^{\pi}(s, \mathbf{a}) = \widehat{r}_{h,j}(s, \mathbf{a}) + \left\langle \widehat{P}_h(s, \mathbf{a}), \underline{V}_{h+1,j}^{\pi} \right\rangle$

    Set $\underline{V}_{h,j}^{\pi}(s) = \text{proj}_{[0,H-h+1]} \left\{ \mathbb{E}_{\mathbf{a} \sim \pi_h(\cdot|s)} \underline{Q}_{h,j}^{\pi}(s, \mathbf{a}) - b_h(s, \pi_h^s) \right\}$

    Set $\overline{Q}_{h,j}^{\pi}(s, \mathbf{a}) = \widehat{r}_{h,j}(s, \mathbf{a}) + \left\langle \widehat{P}_h(s, \mathbf{a}), \overline{V}_{h+1,j}^{\pi} \right\rangle + H\mathbf{1}\{\mathbf{a} \notin \mathcal{K}_h(s)\}$

    Set $\overline{V}_{h,j}^{\pi}(s) = \text{proj}_{[0,H-h+1]} \left\{ \mathbb{E}_{\mathbf{a} \sim \pi_h(\cdot|s)} \overline{Q}_{h,j}^{\pi}(s, \mathbf{a}) + b_h(s, \pi_h^s) \right\}$

**end for**

Output $\underline{V}_{1,j}^{\pi}(s_1)$ and $\overline{V}_{1,j}^{\pi}(s_1)$.

---

**Algorithm 3** Best Response Estimation

---

Input: offline dataset $\mathcal{D}$, player index $j$ and strategy $\pi_{-j}$.

Initialization: $\overline{V}_{H+1,j}^{*,\pi_{-j}}(s) = 0$ for all $s \in \mathcal{S}$.

**for** time $h = H, H-1, \ldots, 1$ **do**

    Set $\overline{Q}_{h,j}^{*,\pi_{-j}}(s, \mathbf{a}) = \widehat{r}_{h,j}(s, \mathbf{a}) + \left\langle \widehat{P}_h(s, \mathbf{a}), \overline{V}_{h+1,j}^{*,\pi_{-j}} \right\rangle + H\mathbf{1}\{\mathbf{a} \notin \mathcal{K}_h(s)\}$

    Set $\overline{V}_{h,j}(s, a_j) = \mathbb{E}_{\mathbf{a}_{-j} \sim \pi_{h,-j}(\cdot|s)} \overline{Q}_{h,j}^{*,\pi_{-j}}(s, \mathbf{a}) + b_h(s, a_j, \pi_{h,-j}^s)$

    Set $\overline{V}_{h,j}^{*,\pi_{-j}}(s) = \text{proj}_{[0,H-h+1]} \left\{ \max_{a_j \in \mathcal{A}_j} \overline{V}_{h,j}(s, a_j) \right\}$

**end for**

Output $\overline{V}_{1,j}^{*,\pi_{-j}}(s_1)$.

---

**Algorithm 4** **S**trategy-wise **B**onus + **S**urrogate **M**inimization (SBSM)

---

Input: offline dataset $\mathcal{D}$.

$\pi^{\text{output}} = \text{argmin}_{\pi \in \Pi} \sum_{j \in [m]} \overline{V}_{1,j}^{*,\pi_{-j}}(s_1) - \underline{V}_{1,j}^{\pi}(s_1)$, where $\overline{V}_{1,j}^{*,\pi_{-j}}(s_1)$ and $\underline{V}_{1,j}^{\pi}(s_1)$ are computed via Algorithm 3 and Algorithm 2.

Output $\pi^{\text{output}}$.

---

# B Technical Lemmas

## B.1 Covering Number of Strategy Classes

**Lemma 1.** *For the no prior knowledge setting ($\Pi = \Pi^{\text{full}}$), we have*

$$\log \mathcal{N}(\Pi) = \widetilde{O}\left(\sum_{j \in [m]} A_j \log(1/\epsilon_{\text{cover}})\right).$$

*Proof.* If $\Pi = \Pi^{\text{full}}$, by Lemma 28 we have

$$\log \mathcal{N}(\Pi) = \log\left(\sum_{s \in \mathcal{S}, h \in [H]} \prod_{j \in [m]} |\mathcal{C}(\Pi_{h,j}(s), \epsilon_{\text{cover}})|\right)$$

$$= \log\left(SH \prod_{j \in [m]} |\mathcal{C}(\Delta(\mathcal{A}_j), \epsilon_{\text{cover}})|\right)$$

$$\begin{aligned}
&= \sum_{j \in [m]} \log\left(\mathcal{C}(\Delta(\mathcal{A}_j), \epsilon_{\text{cover}})\right) + \log(SH) \\
&\leq \sum_{j \in [m]} A_j \log\left(3A_j / \epsilon_{\text{cover}}\right) + \log(SH) \qquad \text{(Lemma 28)} \\
&= \widetilde{O}\Big(\sum_{j \in [m]} A_j \log(1/\epsilon_{\text{cover}})\Big).
\end{aligned}$$

$\qquad\square$

**Lemma 2.** *If $\Pi$ is a finite set, we have*

$$\log(\mathcal{N}(\Pi)) \leq m \log(|\Pi|) + \log(SH).$$

*Proof.* We have $|\mathcal{C}(\Pi_{h,j}(s), \epsilon_{\text{cover}})| \leq |\Pi_{h,j}(s)| \leq |\Pi|$ for all $h \in [H]$ and $j \in [m]$. Plug it into the definition of $\mathcal{N}(\Pi)$ and we can prove the argument. $\qquad\square$

### B.2 Convexity in Two-player Zero-sum Games

In this section, we prove that $\underline{V}_h^{\mu_h^s, \nu_h^s}(s)$ is concave and $\overline{V}_h^{\mu_h^s, \nu_h^s}(s)$ is convex for both $\mu_h^s$ and $\nu_h^s$. In addition, we show that (10) and (11) can be achieved efficiently.

**Lemma 3.** *For any coefficient $c(a_i, b_j)$ s.t. $c(a_i, b_j) \geq 0$ for all $a_i \in \mathcal{A}$ and $b_j \in \mathcal{B}$, function $f(\mu, \nu) = \sqrt{\sum_{a_i \in \mathcal{A}, b_j \in \mathcal{B}} c(a_i, b_j)\mu(a_i)^2 \nu(b_j)^2}$ defined on $\mu \in \Delta(\mathcal{A})$ and $\nu \in \Delta(\mathcal{B})$ is a convex function and $\sqrt{\sum_{a_i \in \mathcal{A}} \sum_{b_j \in \mathcal{B}} c(a_i, b_j)}$-Lipschitz continuous function with respect to $\nu$. In addition, it is convex and $\sqrt{\sum_{a_i \in \mathcal{A}} \sum_{b_j \in \mathcal{B}} c(a_i, b_j)}$-Lipschitz continuous with respect to $\mu$ by symmetry.*

*Proof.* We use the convention that $\frac{0}{0} = 0$. We first compute the first-order derivatives

$$\frac{\partial f}{\partial \nu(b_j)} = \frac{\sum_{a_i \in \mathcal{A}} c(a_i, b_j)\mu(a_i)\nu(b_j)^2}{\sqrt{\sum_{a_i \in \mathcal{A}, b \in \mathcal{B}} c(a_i, b)\mu(a_i)^2 \nu(b)^2}}. \tag{18}$$

By Cauchy-Schwarz inequality, we have

$$\begin{aligned}
\frac{\partial f}{\partial \nu(b_j)} &= \frac{\sum_{a_i \in \mathcal{A}} c(a_i, b_j)\mu(a_i)\nu(b_j)^2}{\sqrt{\sum_{a_i \in \mathcal{A}, b \in \mathcal{B}} c(a_i, b)\mu(a_i)^2 \nu(b)^2}} \\
&\leq \frac{\sum_{a_i \in \mathcal{A}} c(a_i, b_j)\mu(a_i)\nu(b_j)}{\sqrt{\sum_{a_i \in \mathcal{A}} c(a_i, b_j)\mu(a_i)^2 \nu(b_j)^2}} \\
&\leq \sqrt{\sum_{a_i \in \mathcal{A}} c(a_i, b_j)}.
\end{aligned}$$

Then we have

$$\left\|\frac{\partial f}{\partial \nu}\right\|_2 \leq \sqrt{\sum_{a_i \in \mathcal{A}} \sum_{b_j \in \mathcal{B}} c(a_i, b_j)},$$

which implies $f(\mu, \cdot)$ is $\sqrt{\sum_{a_i \in \mathcal{A}} \sum_{b_j \in \mathcal{B}} c(a_i, b_j)}$-Lipschitz continuous.

The second-order derivatives are

$$\frac{\partial^2 f}{\partial \nu(b_j)\partial \nu(b_k)} = -\frac{\left(\sum_{a_i \in \mathcal{A}} c(a_i, b_j)\mu(a_i)\nu(b_j)^2\right) \cdot \left(\sum_{a_i \in \mathcal{A}} c(a_i, b_k)\mu(a_i)\nu(b_k)^2\right)}{\left(\sum_{a_i \in \mathcal{A}, b_j \in \mathcal{B}} c(a_i, b_j)\mu(a_i)^2 \nu(b_j)^2\right)^{3/2}}, j \neq k,$$

$$\frac{\partial^2 f}{(\partial \nu(b_j))^2} = \frac{\sum_{a_i \in \mathcal{A}} c(a_i, b_j)\nu(b_j)^2}{\sqrt{\sum_{a_i \in \mathcal{A}, b_j \in \mathcal{B}} c(a_i, b_j)\mu(a_i)^2\nu(b_j)^2}} - \frac{\left(\sum_{a_i \in \mathcal{A}} c(a_i, b_j)\mu(a_i)\nu(b_j)^2\right)^2}{\left(\sum_{a_i \in \mathcal{A}, b_j \in \mathcal{B}} c(a_i, b_j)\mu(a_i)^2\nu(b_j)^2\right)^{3/2}}.$$

Then for arbitrary $x \in \mathbb{R}^B$, we have

$$\sum_{j,k \in [B]} x_j x_k \frac{\partial^2 f}{\partial \nu(b_j) \partial \nu(b_k)}$$

$$= \sum_{j \in [B]} \frac{x_j^2 \sum_{a_i \in \mathcal{A}} c(a_i, b_j)\nu(b_j)^2}{\sqrt{\sum_{a_i \in \mathcal{A}, b_j \in \mathcal{B}} c(a_i, b_j)\mu(a_i)^2\nu(b_j)^2}}$$

$$- \sum_{j,k \in [B]} \frac{x_j x_k \left(\sum_{a_i \in \mathcal{A}} c(a_i, b_j)\mu(a_i)\nu(b_j)^2\right) \cdot \left(\sum_{a_i \in \mathcal{A}} c(a_i, b_k)\mu(a_i)\nu(b_k)^2\right)}{\left(\sum_{a_i \in \mathcal{A}, b_j \in \mathcal{B}} c(a_i, b_j)\mu(a_i)^2\nu(b_j)^2\right)^{3/2}}$$

$$= \frac{\sum_{j \in [B]} \left(x_j^2 \sum_{a_i \in \mathcal{A}} c(a_i, b_j)\nu(b_j)^2\right) \cdot \sum_{a_i \in \mathcal{A}, b_j \in \mathcal{B}} c(a_i, b_j)\mu(a_i)^2\nu(b_j)^2}{\left(\sum_{a_i \in \mathcal{A}, b_j \in \mathcal{B}} c(a_i, b_j)\mu(a_i)^2\nu(b_j)^2\right)^{3/2}}$$

$$- \frac{\left(\sum_{j \in [B]} x_j \left(\sum_{a_i \in \mathcal{A}} c(a_i, b_j)\mu(a_i)\nu(b_j)^2\right)\right)^2}{\left(\sum_{a_i \in \mathcal{A}, b_j \in \mathcal{B}} c(a_i, b_j)\mu(a_i)^2\nu(b_j)^2\right)^{3/2}}$$

$$= \frac{\sum_{j \in [B]} \left(x_j^2 \sum_{a_i \in \mathcal{A}} c(a_i, b_j)\nu(b_j)^2\right) \cdot \sum_{a_i \in \mathcal{A}, b_j \in \mathcal{B}} c(a_i, b_j)\mu(a_i)^2\nu(b_j)^2}{\left(\sum_{a_i \in \mathcal{A}, b_j \in \mathcal{B}} c(a_i, b_j)\mu(a_i)^2\nu(b_j)^2\right)^{3/2}}$$

$$- \frac{\left(\sum_{j \in [B]} x_j \left(\sum_{a_i \in \mathcal{A}} c(a_i, b_j)\mu(a_i)\nu(b_j)^2\right)\right)^2}{\left(\sum_{a_i \in \mathcal{A}, b_j \in \mathcal{B}} c(a_i, b_j)\mu(a_i)^2\nu(b_j)^2\right)^{3/2}}.$$

By Cauchy-Schwarz's inequality, we have

$$\sum_{j \in [B]} \left(x_j^2 \sum_{a_i \in \mathcal{A}} c(a_i, b_j)\nu(b_j)^2\right) \cdot \sum_{a_i \in \mathcal{A}, b_j \in \mathcal{B}} c(a_i, b_j)\mu(a_i)^2\nu(b_j)^2$$

$$= \left(\sum_{j \in [B]} x_j^2 \nu(b_j)^2 \sum_{a_i \in \mathcal{A}} c(a_i, b_j)\right) \cdot \left(\sum_{j \in [B]} \nu(b_j)^2 \sum_{a_i \in \mathcal{A}} c(a_i, b_j)\mu(a_i)^2\right)$$

$$\geq \left(\sum_{j \in [B]} x_j \nu(b_j)^2 \sqrt{\sum_{a_i \in \mathcal{A}} c(a_i, b_j) \sum_{a_i \in \mathcal{A}} c(a_i, b_j)\mu(a_i)^2}\right)^2$$

$$\geq \left(\sum_{j \in [B]} x_j \nu(b_j)^2 \sum_{a_i \in \mathcal{A}} c(a_i, b_j)\mu(a_i)\right)^2$$

$$\geq 0.$$

Thus for arbitrary $x \in \mathbb{R}^B$, we have

$$\sum_{j,k \in [B]} x_j x_k \frac{\partial^2 f}{\partial \nu(b_j) \partial \nu(b_k)} \geq 0,$$

$\square$

which implies $f$ is convex with respect to $\nu$.

**Proposition 6.** *For all $h \in [H]$ and $s \in \mathcal{S}$, $\underline{V}_h^{\mu_h^s, \nu_h^s}$ is concave and $H + H\sqrt{\log(\mathcal{N}(\Pi))\iota}$-Lipschitz with respect to $\mu_h^s$ and $\nu_h^s$. Similarly, $\overline{V}_h^{\mu_h^s, \nu_h^s}$ is convex with respect to $\mu_h^s$ and $\nu_h^s$. As a result, (10) and (11) can be achieved with $(H + H\sqrt{\log(\mathcal{N}(\Pi))\iota})^2/\epsilon_{\mathrm{opt}}^2$ iterations by projected gradient descent.*

*Proof.* Recall that

$$\underline{V}_h^{\mu_h^s, \nu_h^s}(s) = \mathbb{E}_{a \sim \mu_h^s, b \sim \nu_h^s} \underline{Q}_h(s, a, b) - H\sqrt{\sum_{(a,b) \in \mathcal{K}_h(s)} \frac{\mu_h^s(a)^2 \nu_h^s(b)^2}{n_h(s, a, b)} \log(\mathcal{N}(\Pi))\iota} - \sqrt{\iota}/n.$$

The first term is linear with respect to $\mu_h^s$, The second term is convex by Lemma 3 and the last term is a constant. As a result, $\underline{V}_h^{\mu_h^s, \nu_h^s}$ is concave with respect to $\mu_h^s$. By symmetry, it is also concave with respect to $\nu_h^s$. The proof for $\overline{V}_h^{\mu_h^s, \nu_h^s}$ is the same. The Lipschitz constant is a direct implication of Lemma 3. The iteration complexity of projected gradient descent is from Section 3.1 in Bubeck et al. [2015]. Note that in each iteration we only need to compute the gradient (18) and a projection onto the probability simplex. $\qquad\square$

### B.3 Convexity in Multi-player General-sum Games

In this section, we will show that the bonus $b_h(s, \pi_h^s)$ in multi-player general-sum game is also convex with respect to $\pi_{h,j}^s$ for all $j \in [m]$.

**Lemma 4.** *For any $h \in [H]$ and $s \in \mathcal{S}$, $b_h(s, \pi_h^s)$ is convex with respect to $\pi_{h,j}^s$.*

*Proof.* Recall that

$$b_h(s, \pi_h^s) = H\sqrt{\sum_{\mathbf{a} \in \mathcal{K}_h(s)} \frac{\pi_h^s(\mathbf{a})^2}{n_h(s, \mathbf{a})} \log(\mathcal{N}(\Pi))\iota} + \sqrt{\iota}/n.$$

As we have

$$\sum_{\mathbf{a} \in \mathcal{K}_h(s)} \frac{\pi_h^s(\mathbf{a})^2}{n_h(s, \mathbf{a})} = \sum_{a_j \in \mathcal{A}_j} \sum_{\mathbf{a}_{-j}:(a_j, \mathbf{a}_{-j}) \in \mathcal{K}_h(s)} \frac{\pi_{h,j}^s(a_j)^2 \pi_{h,-j}^s(\mathbf{a}_{-j})^2}{n_h(s, \mathbf{a})},$$

by Lemma 3 we have that $b_h(s, \pi_h^s)$ is convex with respect to $\pi_{h,j}^s$. $\qquad\square$

One direction implication is that $\max_{\pi_{h,j}^s} \overline{V}_{h,j}^\pi(s)$ can be achieved by a deterministic strategy $\pi_{h,j}^s \in D(\mathcal{A}_j)$, which will be utilized in Appendix D.

## C  Proofs in Section 3

**Lemma 5.** *Fix $h \in [H]$ and $s \in \mathcal{S}$, $\mu_h'(\cdot|s) \in \Delta(\mathcal{A})$, $\nu_h'(\cdot|s) \in \Delta(\mathcal{B})$, with probability $1 - \delta$ we have*

$$\left| \sum_{(a,b) \in \mathcal{K}_h(s)} \mu_h'(a|s) \nu_h'(b|s) \left( r_h(s, a, b) + \langle P_h(s, a, b), \underline{V}_{h+1} \rangle - \widehat{r}_h(s, a, b) - \langle \widehat{P}_h(s, a, b), \underline{V}_{h+1} \rangle \right) \right|$$

$$\leq H\sqrt{2 \sum_{(a,b) \in \mathcal{K}_h(s)} \frac{\mu_h'(a|s)^2 \nu_h'(b|s)^2}{n_h(s, a, b)} \log(2/\delta)},$$

$$\left| \sum_{(a,b) \in \mathcal{K}_h(s)} \mu_h'(a|s) \nu_h'(b|s) \left( r_h(s, a, b) + \langle P_h(s, a, b), \overline{V}_{h+1} \rangle - \widehat{r}_h(s, a, b) - \langle \widehat{P}_h(s, a, b), \overline{V}_{h+1} \rangle \right) \right|$$

$$\leq H\sqrt{2 \sum_{(a,b) \in \mathcal{K}_h(s)} \frac{\mu_h'(a|s)^2 \nu_h'(b|s)^2}{n_h(s, a, b)} \log(2/\delta)}.$$

*Proof.* We use $k_h^i(s,a,b)$ to denote the index of $(s,a,b)$ appears in the dataset at timestep $h$ for $i$th time. We prove the first argument and the second argument holds similarly. With probability $1-\delta$, we have

$$\left| \sum_{(a,b)\in\mathcal{K}_h(s)} \mu_h'(a|s)\nu_h'(b|s)\left( r_h(s,a,b) + \langle P_h(s,a,b),\underline{V}_{h+1}\rangle - \widehat{r}_h(s,a,b) - \left\langle \widehat{P}_h(s,a,b),\underline{V}_{h+1}\right\rangle\right)\right|$$

$$= \left| \sum_{(a,b)\in\mathcal{K}_h(s)} \sum_{i=1}^{n_h(s,a,b)} \frac{\mu_h'(a|s)\nu_h'(b|s)}{n_h(s,a,b)}\left( r_h^{k_h^i(s,a,b)} - r_h(s,a,b)\right)\right.$$

$$\left. + \sum_{(a,b)\in\mathcal{K}_h(s)} \sum_{i=1}^{n_h(s,a,b)} \frac{\mu_h'(a|s)\nu_h'(b|s)}{n_h(s,a,b)}\left( \underline{V}_{h+1}(s_{h+1}^{k_h^i(s,a,b)}) - \langle P_h(s,a,b),\underline{V}_{h+1}\rangle\right)\right|$$

$$\leq \sqrt{\frac{1}{2}\sum_{(a,b)\in\mathcal{K}_h(s)} \frac{\mu_h'(a|s)^2\nu_h'(b|s)^2}{n_h(s,a,b)}\log(2/\delta)} + H\sqrt{\frac{1}{2}\sum_{(a,b)\in\mathcal{K}_h(s)} \frac{\mu_h'(a|s)^2\nu_h'(b|s)^2}{n_h(s,a,b)}\log(2/\delta)}$$

$$\leq H\sqrt{2\sum_{(a,b)\in\mathcal{K}_h(s)} \frac{\mu_h'(a|s)^2\nu_h'(b|s)^2}{n_h(s,a,b)}\log(2/\delta)},$$

where the first inequality is from Hoeffding's inequality and the fact that $\underline{V}_{h+1}$ has no dependence on the dataset at timestep $h$. $\qquad\square$

**Lemma 6.** *With probability $1-\delta$, for all $h\in[H], s\in\mathcal{S}, \mu_h^s\in\Pi_h^{\max}(s), \nu_h^s\in D(\mathcal{B})$, we have*

$$\left| \sum_{(a,b)\in\mathcal{K}_h(s)} \mu_h^s(a)\nu_h^s(b)\left( r_h(s,a,b) + \langle P_h(s,a,b),\underline{V}_{h+1}\rangle - \widehat{r}_h(s,a,b) - \left\langle \widehat{P}_h(s,a,b),\underline{V}_{h+1}\right\rangle\right)\right|$$

$$\leq b_h(s,\mu_h^s,\nu_h^s),$$

*and for $\mu_h^s\in D(\mathcal{A}), \nu_h^s\in\Pi_h^{\min}(s)$, we have*

$$\left| \sum_{(a,b)\in\mathcal{K}_h(s)} \mu_h^s(a)\nu_h^s(b)\left( r_h(s,a,b) + \langle P_h(s,a,b),\overline{V}_{h+1}\rangle - \widehat{r}_h(s,a,b) - \left\langle \widehat{P}_h(s,a,b),\overline{V}_{h+1}\right\rangle\right)\right|$$

$$\leq b_h(s,\mu_h^s,\nu_h^s).$$

*Denote this event as $\mathcal{G}$.*

*Proof.* We prove the first argument and the second argument holds similarly. First, using a union bound for all $h\in[H], s\in\mathcal{S}, \mu_h'^s\in\mathcal{C}(\Pi_h^{\max}(s)), \nu_h'^s\in D(\mathcal{B})$ on Lemma 5, with probability $1-\delta$, we have

$$\left| \sum_{(a,b)\in\mathcal{K}_h(s)} \mu_h'^s(a)\nu_h'^s(b)\left( r_h(s,a,b) + \langle P_h(s,a,b),\underline{V}_{h+1}\rangle - \widehat{r}_h(s,a,b) - \left\langle \widehat{P}_h(s,a,b),\underline{V}_{h+1}\right\rangle\right)\right|$$

$$\leq H\sqrt{2\sum_{(a,b)\in\mathcal{K}_h(s)} \frac{\mu_h'^s(a)^2\nu_h'^s(b)^2}{n_h(s,a,b)}\log(2\sum_{s\in\mathcal{S},h\in[H]}|(\mathcal{C}(\Pi_h^{\max}(s))|B + |\mathcal{C}(\Pi_h^{\min}(s))|A)/\delta)}$$

$$\leq H\sqrt{2\sum_{(a,b)\in\mathcal{K}_h(s)} \frac{\mu_h'^s(a)^2\nu_h'^s(b)^2}{n_h(s,a,b)}\log(2\mathcal{N}(\Pi)ABSH\delta)}. \qquad\text{(See Definition 1)}$$

Note that $r_h(s,a,b)+\langle P_h(s,a,b),\underline{V}_{h+1}\rangle-\widehat{r}_h(s,a,b)-\left\langle \widehat{P}_h(s,a,b),\underline{V}_{h+1}\right\rangle$ is bounded in $[-H,H]$ as $r_h(s,a,b)\in[0,1]$ and $\underline{V}_{h+1}\in[0,H-h]$. For any $\mu_h(\cdot|s)\in\Pi_h^{\max}(s)$ and $\nu_h(\cdot|s)\in D(\mathcal{B})$,

there exists $\mu'_h(\cdot|s) \in \mathcal{C}(\Pi_h^{\max}(s))$ and $\nu'_h(\cdot|s) \in D(\mathcal{B})$ such that $\|\mu_h(\cdot|s) - \mu'_h(\cdot|s)\| \leq \epsilon_{\text{cover}}$ and $\|\nu_h(\cdot|s) - \nu'_h(\cdot|s)\| = 0 \leq \epsilon_{\text{cover}}$. So with Lemma 29, we have

$$\left| \sum_{(a,b)\in\mathcal{K}_h(s)} \mu'_h(a|s)\nu'_h(b|s)\left( r_h(s,a,b) + \left\langle P_h(s,a,b), \underline{V}_{h+1}\right\rangle - \widehat{r}_h(s,a,b) - \left\langle \widehat{P}_h(s,a,b), \underline{V}_{h+1}\right\rangle\right) \right.$$

$$\left. - \sum_{(a,b)\in\mathcal{K}_h(s)} \mu_h(a|s)\nu_h(b|s)\left( r_h(s,a,b) + \left\langle P_h(s,a,b), \underline{V}_{h+1}\right\rangle - \widehat{r}_h(s,a,b) - \left\langle \widehat{P}_h(s,a,b), \underline{V}_{h+1}\right\rangle\right) \right|$$

$$\leq 2\epsilon_{\text{cover}} H.$$

By Lemma 30, we have

$$\left| \sqrt{\sum_{(a,b)\in\mathcal{K}_h(s)} \frac{\mu'_h(a|s)^2\nu'_h(b|s)^2}{n_h(s,a,b)}} - \sqrt{\sum_{(a,b)\in\mathcal{K}_h(s)} \frac{\mu_h(a|s)^2\nu_h(b|s)^2}{n_h(s,a,b)}} \right| \leq 2\sqrt{\epsilon_{\text{cover}}}.$$

Combining all these parts together and then with probability $1 - \delta$, we have

$$\left| \sum_{(a,b)\in\mathcal{K}_h(s)} \mu_h(a|s)\nu_h(b|s)\left( r_h(s,a,b) + \left\langle P_h(s,a,b), \underline{V}_{h+1}\right\rangle - \widehat{r}_h(s,a,b) - \left\langle \widehat{P}_h(s,a,b), \underline{V}_{h+1}\right\rangle\right) \right|$$

$$\leq H\sqrt{2\sum_{(a,b)\in\mathcal{K}_h(s)} \frac{\mu_h(a|s)^2\nu_h(b|s)^2}{n_h(s,a,b)} \log(2\mathcal{N}(\Pi,\epsilon_{\text{cover}})ABSH/\delta)} + 2\epsilon_{\text{cover}} H$$

$$+ 2H\sqrt{2\epsilon_{\text{cover}}\log(2\mathcal{N}(\Pi,\epsilon_{\text{cover}})ABSH/\delta)}.$$

Set $\epsilon_{\text{cover}} = \frac{1}{(A+B)H^2n^2}$ and with some algebra we can get

$$\left| \sum_{(a,b)\in\mathcal{K}_h(s)} \mu_h(a|s)\nu_h(b|s)\left( r_h(s,a,b) + \left\langle P_h(s,a,b), \underline{V}_{h+1}\right\rangle - \widehat{r}_h(s,a,b) - \left\langle \widehat{P}_h(s,a,b), \underline{V}_{h+1}\right\rangle\right) \right|$$

$$\leq H\sqrt{2\sum_{(a,b)\in\mathcal{K}_h(s)} \frac{\mu_h(a|s)^2\nu_h(b|s)^2}{n_h(s,a,b)} \log(2\mathcal{N}(\Pi)ABSHn/\delta)} + \sqrt{32\log(2ABSHn/\delta)}/n$$

$$\leq H\sqrt{\sum_{(a,b)\in\mathcal{K}_h(s)} \frac{\mu_h(a|s)^2\nu_h(b|s)^2}{n_h(s,a,b)} \log(\mathcal{N}(\Pi))\iota} + \sqrt{\iota}/n,$$

where $\iota = 32\log(2ABSHn/\delta)$. $\qquad\qquad\square$

**Lemma 7.** *Under event $\mathcal{G}$, for all $s \in \mathcal{S}$, $h \in [H]$, $\mu_h(\cdot|s) \in \Pi_h^{\max}(s)$ and $\nu_h(\cdot|s) \in D(\mathcal{B})$, we have*

$$\underline{V}_h^{\mu_h^s, \nu_h^s}(s) \leq \mathbb{E}_{a\sim\mu_h(\cdot|s), b\sim\nu_h(\cdot|s)}\left[ r_h(s,a,b) + \left\langle P_h(s,a,b), \underline{V}_{h+1}\right\rangle\right],$$

*and for $\mu_h^s \in D(\mathcal{A})$, $\nu_h^s \in \Pi_h^{\min}(s)$, we have*

$$\overline{V}_h^{\mu_h^s, \nu_h^s}(s) \geq \mathbb{E}_{a\sim\mu_h(\cdot|s), b\sim\nu_h(\cdot|s)}\left[ r_h(s,a,b) + \left\langle P_h(s,a,b), \overline{V}_{h+1}\right\rangle\right].$$

*Proof.* Under the good event $\mathcal{G}$, we have

$$\underline{V}_h^{\mu_h^s, \nu_h^s}(s)$$

$$= \mathbb{E}_{a\sim\mu_h(\cdot|s), b\sim\nu_h(\cdot|s)}\underline{Q}_h(s,a,b) - b_h(s, \mu_h^s, \nu_h^s)$$

$$= \sum_{(a,b)\in\mathcal{K}_h(s)} \mu_h(a|s)\nu_h(b|s)\left( \widehat{r}_h(s,a,b) + \left\langle \widehat{P}_h(s,a,b), \underline{V}_{h+1}\right\rangle\right) - b_h(s, \mu_h^s, \nu_h^s)$$

$$\leq \sum_{(a,b)\in\mathcal{K}_h(s)} \mu_h(a|s)\nu_h(b|s)\left(r_h(s,a,b)+\left\langle P_h(s,a,b),\underline{V}_{h+1}\right\rangle\right) \qquad \text{(Lemma 6)}$$

$$\leq \sum_{a\in\mathcal{A},b\in\mathcal{B}} \mu_h(a|s)\nu_h(b|s)\left(r_h(s,a,b)+\left\langle P_h(s,a,b),\underline{V}_{h+1}\right\rangle\right) \qquad (\underline{V}_{h+1}\geq 0)$$

$$=\mathbb{E}_{a\sim\mu_h(\cdot|s),b\sim\nu_h(\cdot|s)}\left[r_h(s,a,b)+\left\langle P_h(s,a,b),\underline{V}_{h+1}\right\rangle\right].$$

Similarly we have

$$\overline{V}_h^{\mu_h^s,\nu_h^s}(s)$$

$$=\mathbb{E}_{a\sim\mu_h(\cdot|s),b\sim\nu_h(\cdot|s)}\overline{Q}_h(s,a,b)+b_h(s,\mu_h^s,\nu_h^s)$$

$$=\sum_{(a,b)\in\mathcal{K}_h(s)} \mu_h(a|s)\nu_h(b|s)\left(\widehat{r}_h(s,a,b)+\left\langle \widehat{P}_h(s,a,b),\overline{V}_{h+1}\right\rangle\right)+H\sum_{(a,b)\notin\mathcal{K}_h(s)} \mu_h(a|s)\nu_h(b|s)$$

$$+b_h(s,\mu_h^s,\nu_h^s)$$

$$\geq \sum_{(a,b)\in\mathcal{K}_h(s)} \mu_h(a|s)\nu_h(b|s)\left(r_h(s,a,b)+\left\langle P_h(s,a,b),\overline{V}_{h+1}\right\rangle\right)+H\sum_{(a,b)\notin\mathcal{K}_h(s)} \mu_h(a|s)\nu_h(b|s)$$

$$\qquad\qquad\qquad\qquad\qquad\qquad\qquad\qquad\qquad\qquad\qquad\qquad\qquad \text{(Lemma 6)}$$

$$\geq \sum_{a\in\mathcal{A},b\in\mathcal{B}} \mu_h(a|s)\nu_h(b|s)\left(r_h(s,a,b)+\left\langle P_h(s,a,b),\overline{V}_{h+1}\right\rangle\right) \qquad (\overline{V}_{h+1}\leq H-h)$$

$$=\mathbb{E}_{a\sim\mu_h(\cdot|s),b\sim\nu_h(\cdot|s)}\left[r_h(s,a,b)+\left\langle P_h(s,a,b),\overline{V}_{h+1}\right\rangle\right].$$

$\square$

**Lemma 8.** *Under event $\mathcal{G}$, for all $s\in\mathcal{S}$ and $h\in[H]$, with probability $1-\delta$, we have*

$$\underline{V}_h(s)\leq V_h^{\mu,*}(s),\overline{V}_h(s)\geq V_h^{*,\overline{\nu}}(s).$$

*Proof.* We prove the first argument and the second argument holds similarly. We prove this argument by induction. It holds trivially for $h=H+1$ as both sides are equal to zero. Suppose the argument holds for timestep $h+1$. Then for any $s\in\mathcal{S}$, we have

$$\underline{V}_h(s)=\mathrm{proj}_{[0,H-h+1]}\left\{\underline{V}_h^{\mu_h^s,\nu_h^s}(s)\right\}$$

$$=\mathrm{proj}_{[0,H-h+1]}\left\{\min_{\nu_h^s\in D(\mathcal{B})} \underline{V}_h^{\mu_h^s,\nu_h^s}(s)\right\}$$

$$\leq \mathrm{proj}_{[0,H-h+1]}\left\{\min_{\nu_h^s\in D(\mathcal{B})} \mathbb{E}_{a\sim\mu_h(\cdot|s),b\sim\nu_h(\cdot|s)}\left[r_h(s,a,b)+\left\langle P_h(s,a,b),\underline{V}_{h+1}\right\rangle\right]\right\}$$

$$\qquad\qquad\qquad\qquad\qquad\qquad\qquad\qquad\qquad\qquad\qquad\qquad\qquad \text{(Lemma 7)}$$

$$\leq \mathrm{proj}_{[0,H-h+1]}\left\{\min_{\nu_h^s\in D(\mathcal{B})} \mathbb{E}_{a\sim\mu_h(\cdot|s),b\sim\nu_h(\cdot|s)}\left[r_h(s,a,b)+\left\langle P_h(s,a,b),V_{h+1}^{\mu,*}\right\rangle\right]\right\}$$

$$\qquad\qquad\qquad\qquad\qquad\qquad\qquad\qquad\qquad\qquad\qquad\qquad \text{(Induction hypothesis)}$$

$$=\mathrm{proj}_{[0,H-h+1]}\left\{V_h^{\mu,*}(s)\right\} \qquad \text{(There always exists a best response in } D(\mathcal{B}))$$

$$=V_h^{\mu,*}(s).$$

By induction, the argument holds for all $h\in[H]$. The proof for $\overline{V}_h(s)$ is the same. $\square$

For any $\mu_h^s\in\Delta(\mathcal{A})$, with a slight abuse of notation, we define

$$\underline{\nu}_h^s(\mu_h^s):=\operatorname*{argmin}_{\nu_h^s\in D(\mathcal{B})} \underline{V}_h^{\mu_h^s,\nu_h^s}.$$

Note that $\underline{\nu}_h^s=\underline{\nu}_h^s(\underline{\mu}_h^s)$. We use $\underline{\nu}(\mu)\in\Pi^{\min,\det}$ to denote a strategy for player 2 such that she use $\underline{\nu}_h^s(\mu_h^s)$ at state $s$ and timestep $h$.

**Lemma 9.** *Under the good event $\mathcal{G}$, for any $\widetilde{\mu} \in \Pi^{\max}$ and $\widetilde{\nu} \in \Pi^{\min}$, we have*

$$V_1^{\widetilde{\mu},*}(s_1) - V_1^{\underline{\mu},*}(s_1) \le \mathbb{E}_{\widetilde{\mu},\underline{\nu}(\widetilde{\mu})} \sum_{h=1}^{H} \widehat{b}_h(s_h, \widetilde{\mu}_h^{s_h}, \underline{\nu}_h^{s_h}(\widetilde{\mu}_h^{s_h})) + H\epsilon_{\mathrm{opt}},$$

$$V_1^{*,\overline{\nu}}(s_1) - V_1^{*,\widetilde{\nu}}(s_1) \le \mathbb{E}_{\overline{\mu}(\widetilde{\nu}),\widetilde{\nu}} \sum_{h=1}^{H} \widehat{b}_h(s_h, \overline{\mu}_h^{s_h}(\widetilde{\nu}_h^{s_h}), \widetilde{\nu}_h^{s_h}) + H\epsilon_{\mathrm{opt}}.$$

*Proof.* We prove the first argument and the second argument holds similarly. By Lemma 8, we have

$$V_1^{\widetilde{\mu},*}(s_1) - V_1^{\underline{\mu},*}(s_1) \le V_1^{\widetilde{\mu},*}(s_1) - \underline{V}_1(s_1).$$

Now we work on the difference between the NE value and the pessimistic estimate.

$$V_1^{\widetilde{\mu},*}(s_1) - \underline{V}_1(s_1)$$

$$= \min_{\nu_1^{s_1}} \mathbb{E}_{\widetilde{\mu}_1^{s_1},\nu_1^{s_1}} Q_1^{\widetilde{\mu},*}(s_1,a_1,b_1) - \mathrm{proj}_{[0,H]} \left\{ \underline{V}_1^{\mu_1^{s_1},\nu_1^{s_1}}(s_1) \right\}$$

$$\le \min_{\nu_1^{s_1}} \mathbb{E}_{\widetilde{\mu}_1^{s_1},\nu_1^{s_1}} Q_1^{\widetilde{\mu},*}(s_1,a_1,b_1) - \underline{V}_1^{\mu_1^{s_1},\nu_1^{s_1}}(s_1) \qquad\qquad (\underline{V}_1^{\mu_1^{s_1},\nu_1^{s_1}}(s_1) \le H \text{ by (4) and (5)})$$

$$\le \mathbb{E}_{\widetilde{\mu}_1^{s_1},\nu_1^{s_1}(\widetilde{\mu}_1^{s_1})} Q_1^{\widetilde{\mu},*}(s_1,a_1,b_1) - \underline{V}_1^{\widetilde{\mu}_1^{s_1},\nu_1^{s_1}(\widetilde{\mu}_1)}(s_1) + \epsilon_{\mathrm{opt}}$$

$$= \mathbb{E}_{\widetilde{\mu}_1^{s_1},\nu_1^{s_1}(\widetilde{\mu}_1)} \left[ Q_1^{\widetilde{\mu},*}(s_1,a_1,b_1) - \underline{Q}_1(s_1,a_1,b_1) \right] + b_1(s_1, \widetilde{\mu}_1^{s_1}, \nu_1^{s_1}(\widetilde{\mu}_1^{s_1})) + \epsilon_{\mathrm{opt}}$$

$$= \mathbb{E}_{\widetilde{\mu}_1^{s_1},\nu_1^{s_1}(\widetilde{\mu}_1^{s_1})} \left[ r_1(s_1,a_1,b_1) + \left\langle P_1(s_1,a_1,b_1), V_2^{\widetilde{\mu},*} \right\rangle - \widehat{r}_1(s_1,a_1,b_1) - \left\langle \widehat{P}_1(s_1,a_1,b_1), \underline{V}_2 \right\rangle \right]$$
$$\quad + b_1(s_1, \widetilde{\mu}_1^{s_1}, \nu_1^{s_1}(\widetilde{\mu}_1^{s_1})) + \epsilon_{\mathrm{opt}}$$

$$\le \mathbb{E}_{\widetilde{\mu}_1^{s_1},\nu_1^{s_1}(\widetilde{\mu}_1^{s_1})} \left[ V_2^{\widetilde{\mu},*}(s_2) - \underline{V}_2(s_2) \right] + 2b_1(s_1, \widetilde{\mu}_1^{s_1}, \nu_1^{s_1}(\widetilde{\mu}_1^{s_1}))$$
$$\quad + H \sum_{(a_1,b_1) \notin \mathcal{K}_1(s_1)} \widetilde{\mu}_1^{s_1}(a_1) \nu_1^{s_1}(\widetilde{\mu}_1^{s_1})(b_1) + \epsilon_{\mathrm{opt}} \qquad\qquad \text{(Lemma 6)}$$

$$\le \mathbb{E}_{\widetilde{\mu},\underline{\nu}(\widetilde{\mu})} \sum_{h=1}^{H} \left( 2b_h(s_h, \widetilde{\mu}_h^{s_h}, \underline{\nu}_h^{s_h}(\widetilde{\mu}_h^{s_h})) + H \sum_{(a_h,b_h) \notin \mathcal{K}_h(s_h)} \widetilde{\mu}_h^{s_h}(a_h) \underline{\nu}_h^{s_h}(\widetilde{\mu}_h^{s_h})(b_h) \right) + H\epsilon_{\mathrm{opt}},$$

where the last inequality is from telescoping from $h=1$ to $h=H$. $\qquad\square$

**Proposition 7.** *Under the good event $\mathcal{G}$, we have*

$$\mathrm{Gap}(\pi^{\mathrm{output}}) \le$$

$$\min_{\pi=(\mu,\nu) \in \Pi} \max_{\pi'=(\mu',\nu') \in \Pi^{\mathrm{det}}} \left[ \mathrm{Gap}(\pi) + \mathbb{E}_{\mu,\nu'} \sum_{h=1}^{H} \widehat{b}_h(s_h, \mu_h^{s_h}, \nu_h'^{s_h}) + \mathbb{E}_{\mu',\nu} \sum_{h=1}^{H} \widehat{b}_h(s_h, \mu_h'^{s_h}, \nu_h^{s_h}) \right] + 2H\epsilon_{\mathrm{opt}}.$$

*Proof.* This is a direct deduction of Lemma 9. Note that $(\underline{\nu}(\widetilde{\mu}), \overline{\mu}(\widetilde{\nu})) \in \Pi^{\mathrm{det}}$. $\qquad\square$

## C.1 Dataset-dependent Bound

**Lemma 10.** *Suppose $\widehat{C}(\mu,\nu)$ is finite. For any $h \in [H]$ and strategy $\mu'$ and $\nu'$, we have*

$$\mathbb{E}_{\mu,\nu'} b_h(s_h, \mu_h^{s_h}, \nu_h'^{s_h}) \le 2H\sqrt{S\log(\mathcal{N}(\Pi))\widehat{C}(\mu,\nu)\iota/n},$$

$$\mathbb{E}_{\mu',\nu} b_h(s_h, \mu_h'^{s_h}, \nu_h^{s_h}) \le 2H\sqrt{S\log(\mathcal{N}(\Pi))\widehat{C}(\mu,\nu)\iota/n}.$$

*Proof.* We prove the first argument and the second argument holds similarly.

$$\mathbb{E}_{\mu,\nu'} b_h(s_h, \mu_h^{s_h}, \nu_h'^{s_h})$$

$$= \mathbb{E}_{\mu,\nu'} \left[ H \sqrt{\sum_{(a,b) \in \mathcal{K}_h(s)} \frac{\mu_h^{s_h}(a)^2 \nu_h'^{s_h}(b)^2}{n_h(s,a,b)} \log(\mathcal{N}(\Pi))\iota} + \frac{\sqrt{\iota}}{n} \right]$$

$$= \sum_{s_h \in \mathcal{S}} H \sqrt{\log(\mathcal{N}(\Pi))\iota} \sqrt{\sum_{(a_h,b_h) \in \mathcal{K}_h(s_h)} \frac{d_h^{\mu,\nu'}(s_h,a_h,b_h)^2}{n_h(s_h,a_h,b_h)}} + \frac{\sqrt{\iota}}{n}$$

$$= \sum_{s_h \in \mathcal{S}} H \sqrt{\log(\mathcal{N}(\Pi))\iota} \sqrt{\sum_{(a_h,b_h) \in \mathcal{K}_h(s_h)} \frac{d_h^{\mu,\nu'}(s_h,a_h,b_h)^2}{n \cdot \widehat{d}_h(s_h,a_h,b_h)}} + \frac{\sqrt{\iota}}{n}$$

$$\leq \sum_{s_h \in \mathcal{S}} H \sqrt{\log(\mathcal{N}(\Pi))\iota} \sqrt{\sum_{(a_h,b_h) \in \mathcal{K}_h(s_h)} d_h^{\mu,\nu'}(s_h,a_h,b_h)\widehat{C}(\mu,\nu)/n} + \frac{\sqrt{\iota}}{n}$$

$$\leq H \sqrt{S \log(\mathcal{N}(\Pi))\widehat{C}(\mu,\nu)\iota/n} + \frac{\sqrt{\iota}}{n}$$

$$\leq 2H \sqrt{S \log(\mathcal{N}(\Pi))\widehat{C}(\mu,\nu)\iota/n}.$$

$\square$

**Lemma 11.** *Suppose $\widehat{C}(\mu,\nu)$ is finite. For any $h \in [H]$ and strategy $\mu'$ and $\nu'$, we have*

$$\mathbb{E}_{\mu,\nu'} \sum_{(a_h,b_h) \notin \mathcal{K}_h(s_h)} \mu_h^{s_h}(a_h)\nu_h'^{s_h}(b_h) = 0,$$

$$\mathbb{E}_{\mu',\nu} \sum_{(a_h,b_h) \notin \mathcal{K}_h(s_h)} \mu_h'^{s_h}(a_h)\nu_h^{s_h}(b_h) = 0.$$

*Proof.* We prove the first argument and the second argument holds similarly.

$$\mathbb{E}_{\mu,\nu'} \sum_{(a_h,b_h) \notin \mathcal{K}_h(s_h)} \mu_h^{s_h}(a_h)\nu_h'^{s_h}(b_h)$$

$$= \mathbb{E}_{\mu,\nu'} \sum_{(a_h,b_h):\widehat{d}_h(s_h,a_h,b_h)=0} \mu_h^{s_h}(a_h)\nu_h'^{s_h}(b_h)$$

$$= \sum_{(a_h,b_h):\widehat{d}_h(s_h,a_h,b_h)=0} d_h^{\mu,\nu'}(s_h,a_h,b_h)$$

$$\leq \sum_{(a_h,b_h):\widehat{d}_h(s_h,a_h,b_h)=0} C(\mu,\nu')\widehat{d}_h(s_h,a_h,b_h)$$

$$= 0.$$

$\square$

**Lemma 12.** *For any strategy $(\mu,\nu) \in \Pi$, we have*

$$\max_{\nu' \in \Pi^{\min,\det}} \mathbb{E}_{\mu,\nu'} \sum_{h=1}^H \widehat{b}_h(s_h,\mu_h^{s_h},\nu_h'^{s_h}) + \max_{\mu' \in \Pi^{\max,\det}} \mathbb{E}_{\mu',\nu} \sum_{h=1}^H \widehat{b}_h(s_h,\mu_h'^{s_h},\nu_h^{s_h})$$

$$\leq 4H^2 \sqrt{S \log(|\mathcal{N}(\Pi)|)\widehat{C}(\mu,\nu)\iota/n}.$$

*Proof.* If $\widehat{C}(\mu,\nu)$ is infinite, the argument holds immediately. Otherwise we can prove it by Lemma 10 and Lemma 11. $\square$

**Theorem 5.** *Suppose $\pi^{\text{output}}$ is the output of Algorithm 1. With probability $1 - \delta$, we have*

$$\text{Gap}(\pi^{\text{output}}) \leq \min_{\pi=(\mu,\nu) \in \Pi} \left[ \text{Gap}(\pi) + 4H^2 \sqrt{S \log(\mathcal{N}(\Pi))\widehat{C}(\pi)\iota/n} \right].$$

*Proof.* This can be derived from Lemma 9, Lemma 12 directly. $\square$

## C.2 Dataset-independent Bound

**Lemma 13.** *With probability $1 - \delta$, for all $h, s, a, b$, we have*

$$n_h(s, a, b) \geq \left(1 - \sqrt{\frac{2\log(SABH/\delta)}{np_{\min}}}\right) nd_h(s, a, b).$$

*As a result, if $n \geq \frac{8\log(SABH/\delta)}{p_{\min}}$, for any strategy $\pi$ we have*

$$2C(\pi) \geq \widehat{C}(\pi).$$

*Proof.* For a fixed $s, a, b, h$, for any $\epsilon > 0$ we have

$$\mathbb{P}(n_h(s, a, b) < (1 - \epsilon)nd_h(s, a, b)) \leq \exp\left(-\frac{\epsilon^2 nd_h(s, a, b)}{2}\right) \leq \exp\left(-\frac{\epsilon^2 np_{\min}}{2}\right).$$

With a union bound, we have

$$\mathbb{P}(\exists h, s, a, b : \mathbb{P}(n_h(s, a, b) < (1 - \epsilon)nd_h(s, a, b))) \leq SABH \exp\left(-\frac{\epsilon^2 np_{\min}}{2}\right).$$

The RHS is smaller than $\delta$ if we set

$$\epsilon = \sqrt{\frac{2\log(SABH/\delta)}{np_{\min}}}.$$

If $n \geq \frac{8\log(SABH/\delta)}{p_{\min}}$, we have

$$\widehat{d}_h(s, a, b) = \frac{n_h(s, a, b)}{n} \geq \frac{d_h(s, a, b)}{2}.$$

By Definition 3 and Definition 2, we have

$$2C(\pi) \geq \widehat{C}(\pi).$$

$\square$

The following Lemma is from Lemma A.1 in Xie et al. [2021b]. For completeness we provide a proof here.

**Lemma 14.** *With probability at least $1 - \delta$, for all $h \in [H]$, $s \in \mathcal{S}$, $a \in \mathcal{A}$ and $b \in \mathcal{B}$, we have*

$$n_h(s, a, b) \vee 1 \geq \frac{nd_h(s, a, b)}{\iota}.$$

*Proof.* For fixed $h \in [H]$, $s \in \mathcal{S}$, $a \in \mathcal{A}$ and $b \in \mathcal{B}$, $n_h(s, a, b)$ is a binomial random variable following $\text{Bin}(n, d_h(s, a, b))$. We show that with probability $1 - \delta$, we have

$$n_h(s, a, b) \vee 1 \geq \frac{nd_h(s, a, b)}{8\log(1/\delta)}.$$

If $d_h(s, a, b) \leq 8\log(1/\delta)/n$, the argument holds directly. Otherwise by the multiplicative Chernoff bound, we have

$$P(n_h(s, a, b) < nd_h(s, a, b)/2) \leq \exp(-nd_h(s, a, b)/8) \leq \delta.$$

So with probability $1 - \delta$, we have $n_h(s, a, b) \geq nd_h(s, a, b)/2 \geq nd_h(s, a, b)/8\log(1/\delta)$. Then with union bound we can prove the lemma. $\square$

**Lemma 15.** *With probability $1 - \delta$ for any $h \in [H]$ we have*

$$\mathbb{E}_{\mu,\nu'}b_h(s_h, \mu_h^{s_h}, \nu_h'^{s_h}) \leq 2H\sqrt{S\log(\mathcal{N}(\Pi))C(\mu,\nu)\iota^2/n},$$

$$\mathbb{E}_{\mu',\nu}b_h(s_h, \mu_h'^{s_h}, \nu_h^{s_h}) \leq 2H\sqrt{S\log(\mathcal{N}(\Pi))C(\mu,\nu)\iota^2/n}.$$

*Proof.* From Lemma 14, with probability $1 - \delta$, for all $h, s, a, b$, we have

$$n_h(s, a, b) \vee 1 \geq \frac{nd_h(s, a, b)}{\iota}.$$

For $(a, b) \in \mathcal{K}_h(s)$, we have $n_h(s, a, b) \geq 1$ and thus $n_h(s, a, b) \geq \frac{nd_h(s,a,b)}{\iota}$.

$$\mathbb{E}_{\mu,\nu'} b_h(s_h, \mu_h^{s_h}, \nu_h'^{s_h})$$

$$= \mathbb{E}_{\mu,\nu'} \left[ H \sqrt{\sum_{(a,b)\in\mathcal{K}_h(s)} \frac{\mu_h^{s_h}(a)^2 \nu_h'^{s_h}(b)^2}{n_h(s, a, b)} \log(\mathcal{N}(\Pi))\iota} + \frac{\sqrt{\iota}}{n} \right]$$

$$= \sum_{s_h \in \mathcal{S}} H \sqrt{\log(\mathcal{N}(\Pi))\iota} \sqrt{\sum_{(a_h,b_h)\in\mathcal{K}_h(s_h)} \frac{d_h^{\mu,\nu'}(s_h, a_h, b_h)^2}{n_h(s_h, a_h, b_h)}} + \frac{\sqrt{\iota}}{n}$$

$$= \sum_{s_h \in \mathcal{S}} H \sqrt{\log(\mathcal{N}(\Pi))\iota^2} \sqrt{\sum_{(a_h,b_h)\in\mathcal{K}_h(s_h)} \frac{d_h^{\mu,\nu'}(s_h, a_h, b_h)^2}{n \cdot d_h(s_h, a_h, b_h)}} + \frac{\sqrt{\iota}}{n}$$

$$\leq \sum_{s_h \in \mathcal{S}} H \sqrt{\log(\mathcal{N}(\Pi))\iota^2} \sqrt{\sum_{(a_h,b_h)\in\mathcal{K}_h(s_h)} d_h^{\mu^*,\underline{\nu}(\mu^*)}(s_h, a_h, b_h)C^*/n} + \frac{\sqrt{\iota}}{n}$$

$$\leq H\sqrt{S\log(\mathcal{N}(\Pi))C^*\iota^2/n} + \frac{\sqrt{\iota}}{n}$$

$$\leq 2H\sqrt{S\log(\mathcal{N}(\Pi))C^*\iota^2/n}.$$

$\square$

**Lemma 16.** *With probability $1 - \delta$ for any $\mu' \in \Pi^{\max,\det}$, $\nu' \in \Pi^{\min,\det}$, $h \in [H]$ and $t \in [0, 1]$ we have*

$$\mathbb{E}_{\mu,\nu'} \sum_{(a_h,b_h)\notin\mathcal{K}_h(s_h)} \mu_h^{s_h}(a_h)\nu_h'^{s_h}(b_h) \leq (SAC(\mu,\nu)\iota/n)^t,$$

$$\mathbb{E}_{\mu',\nu} \sum_{(a_h,b_h)\notin\mathcal{K}_h(s_h)} \mu_h'^{s_h}(a_h)\nu_h^{s_h}(b_h) \leq (SBC(\mu,\nu)\iota/n)^t.$$

*In addition, if $\mu \in \Pi^{\max,\det}$ and $\nu \in \Pi^{\min,\det}$, we have*

$$\mathbb{E}_{\mu,\nu'} \sum_{(a_h,b_h)\notin\mathcal{K}_h(s_h)} \mu_h^{s_h}(a_h)\nu_h'^{s_h}(b_h) \leq (SC(\mu,\nu)\iota/n)^t,$$

$$\mathbb{E}_{\mu',\nu} \sum_{(a_h,b_h)\notin\mathcal{K}_h(s_h)} \mu_h'^{s_h}(a_h)\nu_h^{s_h}(b_h) \leq (SC(\mu,\nu)\iota/n)^t.$$

*Proof.* We prove the first argument and the second one holds similarly. From Lemma 14, with probability $1 - \delta$, for all $h, s, a, b$, we have

$$n_h(s, a, b) \vee 1 \geq \frac{nd_h(s, a, b)}{\iota}.$$

For $(a, b) \notin \mathcal{K}_h(s)$, we have $n_h(s, a, b) = 0$ and thus $\iota \geq nd_h(s, a, b)$. Then for any $t \in [0, 1]$, we have

$$\mathbb{E}_{\mu,\nu'} \sum_{(a_h,b_h)\notin\mathcal{K}_h(s_h)} \mu_h^{s_h}(a_h)\nu_h'^{s_h}(b_h)$$

$$\leq \mathbb{E}_{\mu,\nu'} \sum_{(a_h,b_h)\in\mathcal{A}\times\mathcal{B}} \frac{\mu_h^{s_h}(a_h)\nu_h'^{s_h}(b_h)\iota^t}{(nd_h(s_h, a_h, b_h))^t}$$

$$= \sum_{s_h \in \mathcal{S}} \sum_{a_h\in\mathcal{A},b_h=\nu_h'(s_h)} \frac{d_h^{\mu,\nu'}(s_h, a_h, b_h)\iota^t}{(nd_h(s_h, a_h, b_h))^t}$$

$$\leq \sum_{s_h \in \mathcal{S}} \sum_{a_h \in \mathcal{A}, b_h = \nu'_h(a_h)} \frac{C^t(\mu, \nu)\iota^t}{n^t} \left( d_h^{\mu, \nu'}(s_h, a_h, b_h) \right)^{1-t}$$

$$\leq (SAC(\mu, \nu)\iota/n)^t . \hspace{2cm} \text{(Cauchy-Schwarz Inequality)}$$

If we have $\mu \in M^{\mathrm{det}}$, then we have

$$\mathbb{E}_{\mu, \nu'} \sum_{(a_h, b_h) \notin \mathcal{K}_h(s_h)} \mu_h^{s_h}(a_h) \nu'^{s_h}_h(b_h)$$

$$\leq \sum_{s_h \in \mathcal{S}} \sum_{a_h = \mu_h(s_h), b_h = \nu'_h(s_h)} \frac{C^t(\mu, \nu)\iota^t}{n^t} \left( d_h^{\mu, \nu'}(s_h, a_h, b_h) \right)^{1-t}$$

$$\leq (SC(\mu, \nu)\iota/n)^t . \hspace{2cm} \text{(Cauchy-Schwarz Inequality)}$$

$\square$

**Theorem 6.** *With probability $1 - \delta$, we have*

$$\mathrm{Gap}(\pi^{\mathrm{output}}) \leq \min_{\pi = (\mu, \nu) \in \Pi} \left[ \mathrm{Gap}(\pi) + 4H^2 \sqrt{S \log(\mathcal{N}(\Pi))C(\pi)\iota^2/n} + 2HC(\pi)S(A+B)\iota/n \right].$$

*In additon, if $n \geq \frac{8 \log(SABH/\delta)}{p_{\min}}$, we have*

$$\mathrm{Gap}(\pi^{\mathrm{output}}) \leq \min_{\pi = (\mu, \nu) \in \Pi} \left[ \mathrm{Gap}(\pi) + 8H^2 \sqrt{S \log(\mathcal{N}(\Pi))C(\pi)\iota^2/n} \right].$$

*Proof.* The first argument can be derived by Lemma 15 and Lemma 16 with $t = 1$. The second argument can be derived by Theorem 5 and Lemma 13. $\square$

**Corollary 5.** *If $\Pi = \Pi^{\mathrm{full}}$, then with probability $1 - \delta$ we have*

$$\mathrm{Gap}(\pi^{\mathrm{output}}) = \widetilde{O}(\sqrt{H^4 S(A+B)C(\pi^*)/n}).$$

*In addition, for turn-based two-player zero-sum Markov games, we can set $\Pi = \Pi^{\mathrm{det}}$ and we have*

$$\mathrm{Gap}(\pi^{\mathrm{output}}) = \widetilde{O}(\sqrt{H^4 SC(\pi^*)/n}).$$

*Proof.* The first argument can be derived by Lemma 1 and Theorem 6 with $t = 1/2$. The second argument can be derived by Lemma 2, Lemma 15 and Lemma 16 with $t = 1/2$. $\square$

# D  Proofs in Section 4

**Lemma 17.** *For any strategy $\pi \in \Pi$, $h \in [H]$ and $s_h \in \mathcal{S}$, we have*

$$\overline{V}_{h,j}^{*, \pi_{-j}}(s_h) = \max_{\pi_j} \overline{V}_{h,j}^{\pi}(s_h).$$

*Proof.* We prove this argument by induction. It holds trivially for $H + 1$ as $\overline{V}_{H+1,j}^{*, \pi_{-j}}(s) = \max_{\pi_j} \overline{V}_{H+1,j}^{\pi}(s) = 0$ for any $s \in \mathcal{S}$. Suppose the argument holds for $h + 1$ and now we consider $h$. Consider function

$$f(\pi'^s_{h,j}) = \mathbb{E}_{a_j \sim \pi'^s_{h,j}, \mathbf{a}_{-j} \sim \pi^s_{h,-j}} \widehat{r}_{h,j}(s, a_j, \mathbf{a}_{-j}) + \mathbb{E}_{a_j \sim \pi'^s_{h,j}, \mathbf{a}_{-j} \sim \pi^s_{h,-j}} \widehat{P}_h(s, a_j, \mathbf{a}_{-j}) \cdot \overline{V}_{h+1,j}^{*, \pi_{-j}}$$
$$+ b_h(s, \pi'^s_{h,j}, \pi^s_{h,-j}) + H \sum_{\mathbf{a}_{-j} : (a_j, \mathbf{a}_{-j}) \notin \mathcal{K}(s)} \pi^s_{h,-j}(\mathbf{a}_{-j}).$$

Lemma 4 shows that $b_h(s, \pi'^s_{h,j}, \pi^s_{h,-j})$ is convex with respect to $\pi'^s_{h,j}$, while all the other terms are linear with respect to $\pi'^s_{h,j}$. As a result, $f(\pi'^s_{h,j})$ is a convex function and thus we have

$$\max_{\pi'^s_{h,j} \in \Delta(\mathcal{A}_j)} f(\pi'^s_{h,j}) = \max_{\pi'^s_{h,j} \in D(\mathcal{A}_j)} f(\pi'^s_{h,j}).$$

Then we have

$$\max_{a_j \in \mathcal{A}_j} \overline{V}_{h,j}(s, a_j)$$

$$= \max_{\pi'^s_{h,j} \in D(\mathcal{A}_j)} f(\pi'^s_{h,j})$$

$$= \max_{\pi'^s_{h,j} \in \Delta(\mathcal{A}_j)} f(\pi'^s_{h,j})$$

$$= \max_{\pi'^s_{h,j} \in \Delta(\mathcal{A}_j)} \mathbb{E}_{a_j \sim \pi'^s_{h,j}, \mathbf{a}_{-j} \sim \pi^s_{h,-j}} \widehat{r}_{h,j}(s, a_j, \mathbf{a}_{-j}) + \mathbb{E}_{a_j \sim \pi'^s_{h,j}, \mathbf{a}_{-j} \sim \pi^s_{h,-j}} \widehat{P}_h(s, a_j, \mathbf{a}_{-j}) \cdot \overline{V}^{*,\pi_{-j}}_{h+1,j}$$

$$+ b_h(s, \pi'^s_{h,j}, \pi^s_{h,-j}) + H \sum_{\mathbf{a}_{-j}:(a_j, \mathbf{a}_{-j}) \notin \mathcal{K}(s)} \pi^s_{h,-j}(\mathbf{a}_{-j})$$

$$= \max_{\pi'^s_{h,j} \in \Delta(\mathcal{A}_j)} \mathbb{E}_{a_j \sim \pi'^s_{h,j}, \mathbf{a}_{-j} \sim \pi^s_{h,-j}} \widehat{r}_{h,j}(s, a_j, \mathbf{a}_{-j}) + \max_{\pi_j} \mathbb{E}_{a_j \sim \pi'^s_{h,j}, \mathbf{a}_{-j} \sim \pi^s_{h,-j}} \widehat{P}_h(s, a_j, \mathbf{a}_{-j}) \cdot \overline{V}^{\pi}_{h+1,j}$$

$$+ b_h(s, \pi'^s_{h,j}, \pi^s_{h,-j}) + H \sum_{\mathbf{a}_{-j}:(a_j, \mathbf{a}_{-j}) \notin \mathcal{K}(s)} \pi^s_{h,-j}(\mathbf{a}_{-j}) \qquad \text{(Induction hypothesis)}$$

$$= \max_{\pi_j} \mathbb{E}_{a_j \sim \pi'^s_{h,j}, \mathbf{a}_{-j} \sim \pi^s_{h,-j}} \widehat{r}_{h,j}(s, a_j, \mathbf{a}_{-j}) + \mathbb{E}_{a_j \sim \pi'^s_{h,j}, \mathbf{a}_{-j} \sim \pi^s_{h,-j}} \widehat{P}_h(s, a_j, \mathbf{a}_{-j}) \cdot \overline{V}^{\pi}_{h+1,j}$$

$$+ b_h(s, \pi'^s_{h,j}, \pi^s_{h,-j}) + H \sum_{\mathbf{a}_{-j}:(a_j, \mathbf{a}_{-j}) \notin \mathcal{K}(s)} \pi^s_{h,-j}(\mathbf{a}_{-j}).$$

So we have $\overline{V}^{*,\pi_{-j}}_{h,j}(s_h) = \max_{\pi_j} \overline{V}^{\pi}_{h,j}(s_h)$. (See Algorithm 2 and Algorithm 3 for the definition of both quantities) $\qquad \square$

**Lemma 18.** *Fix* $\pi' \in \Pi, j \in [m], h \in [H]$ *and* $s \in \mathcal{S}$, *with probability* $1 - \delta$ *we have*

$$\left| \sum_{\mathbf{a} \in \mathcal{K}_h(s)} \pi'_h(\mathbf{a}|s) \left( r_{h,j}(s, \mathbf{a}) + \left\langle P_h(s, \mathbf{a}), \underline{V}^{\pi'}_{h+1,j} \right\rangle - \widehat{r}_{h,j}(s, \mathbf{a}) - \left\langle \widehat{P}_{h,j}(s, \mathbf{a}), \underline{V}^{\pi'}_{h+1,j} \right\rangle \right) \right|$$

$$\leq H \sqrt{2 \sum_{\mathbf{a} \in \mathcal{K}_h(s)} \frac{\pi'_h(\mathbf{a}|s)^2}{n_h(s, \mathbf{a})} \log(4/\delta)},$$

*and*

$$\left| \sum_{\mathbf{a} \in \mathcal{K}_h(s)} \pi'_h(\mathbf{a}|s) \left( r_{h,j}(s, \mathbf{a}) + \left\langle P_h(s, \mathbf{a}), \overline{V}^{\pi'}_{h+1,j} \right\rangle - \widehat{r}_{h,j}(s, \mathbf{a}) - \left\langle \widehat{P}_h(s, \mathbf{a}), \overline{V}^{\pi'}_{h+1,j} \right\rangle \right) \right|$$

$$\leq H \sqrt{2 \sum_{\mathbf{a} \in \mathcal{K}_h(s)} \frac{\pi'_h(\mathbf{a}|s)^2}{n_h(s, \mathbf{a})} \log(4/\delta)}.$$

*Proof.* We use $k^i_h(s, a, b)$ to denote the index of $(s, a, b)$ appears in the dataset at timestep $h$ for $i$th time. With probability $1 - \delta$, we have

$$\left| \sum_{(\mathbf{a}) \in \mathcal{K}_h(s)} \pi'_h(\mathbf{a}|s) \left( r_{h,j}(s, \mathbf{a}) + \left\langle P_h(s, \mathbf{a}), \underline{V}^{\pi'}_{h+1,j} \right\rangle - \widehat{r}_{h,j}(s, \mathbf{a}) - \left\langle \widehat{P}_h(s, \mathbf{a}), \underline{V}^{\pi'}_{h+1,j} \right\rangle \right) \right|$$

$$= \left| \sum_{\mathbf{a} \in \mathcal{K}_h(s)} \sum_{i=1}^{n_h(s, \mathbf{a})} \frac{\pi'_h(\mathbf{a}|s)}{n_h(s, \mathbf{a})} \left( r^{k^i_h(s, \mathbf{a})}_{h,j} - r_{h,j}(s, \mathbf{a}) \right) \right.$$

$$\left. + \sum_{(\mathbf{a}) \in \mathcal{K}_h(s)} \sum_{i=1}^{n_h(s, \mathbf{a})} \frac{\pi'_h(\mathbf{a}|s)}{n_h(s, \mathbf{a})} \left( \underline{V}^{\pi'}_{h+1,j}(s^{k^i_h(s, \mathbf{a})}_{h+1}) - \left\langle P_h(s, \mathbf{a}), \underline{V}^{\pi'}_{h+1,j} \right\rangle \right) \right|$$

$$\leq \sqrt{\frac{1}{2} \sum_{\mathbf{a} \in \mathcal{K}_h(s)} \frac{\pi_h'(\mathbf{a}|s)^2}{n_h(s, \mathbf{a})} \log(2/\delta)} + H \sqrt{\frac{1}{2} \sum_{\mathbf{a} \in \mathcal{K}_h(s)} \frac{\pi_h'(\mathbf{a}|s)^2}{n_h(s, \mathbf{a})} \log(2/\delta)}$$

$$\leq H \sqrt{2 \sum_{\mathbf{a} \in \mathcal{K}_h(s)} \frac{\pi_h'(\mathbf{a}|s)^2}{n_h(s, \mathbf{a})} \log(2/\delta)},$$

where the first inequality is from Hoeffding's inequality and the fact that $\underline{V}_{h+1,j}$ has no dependence on the dataset at timestep $h$. The second argument holds similarly. Rescaling $\delta$ to $\delta/2$ and with an union bound we can prove the lemma. $\qquad \square$

**Lemma 19.** *With probability* $1 - \delta$, *for all* $\pi \in \Pi, j \in [m], h \in [H], s \in \mathcal{S}$, *we have*

$$\left| \sum_{\mathbf{a} \in \mathcal{K}_h(s)} \pi_h(\mathbf{a}|s) \left( r_{h,j}(s, \mathbf{a}) + \left\langle P_h(s, \mathbf{a}), \underline{V}_{h+1,j}^\pi \right\rangle - \widehat{r}_{h,j}(s, \mathbf{a}) - \left\langle \widehat{P}_h(s, \mathbf{a}), \underline{V}_{h+1,j}^\pi \right\rangle \right) \right| \leq b_h(s, \pi_h^s),$$

$$\left| \sum_{\mathbf{a} \in \mathcal{K}_h(s)} \pi_h(\mathbf{a}|s) \left( r_{h,j}(s, \mathbf{a}) + \left\langle P_h(s, \mathbf{a}), \overline{V}_{h+1,j}^\pi \right\rangle - \widehat{r}_{h,j}(s, \mathbf{a}) - \left\langle \widehat{P}_h(s, \mathbf{a}), \overline{V}_{h+1,j}^\pi \right\rangle \right) \right| \leq b_h(s, \pi_h^s).$$

*Denote this event as* $\mathcal{G}_{\mathrm{marl}}$.

*Proof.* We prove the argument for $\underline{V}_{h+1,j}^\pi$ and the argument for $\overline{V}_{h+1,j}^\pi$ holds similarly. Suppose $\mathcal{V}$ is a $\epsilon_{\mathrm{cover}}$-covering of $[0, H]^S$ with respect to L-$\infty$ norm and $|\mathcal{V}| \leq (1 + HS/\epsilon_{\mathrm{cover}})^S$. First, using a union bound for all $j \in [m], h \in [H], s \in \mathcal{S}, \pi_{h,j}'^s \in \mathcal{C}(\Pi_{h,j}^{\mathrm{prior}}(s)), V_{h+1} \in \mathcal{V}$ on Lemma 18, with probability $1 - \delta$ we have

$$\left| \sum_{\mathbf{a} \in \mathcal{K}_h(s)} \pi_h'(\mathbf{a}|s) \left( r_{h,j}(s, \mathbf{a}) + \langle P_h(s, \mathbf{a}), V_{h+1} \rangle - \widehat{r}_{h,j}(s, \mathbf{a}) - \left\langle \widehat{P}_h(s, \mathbf{a}), V_{h+1} \right\rangle \right) \right|$$

$$\leq H \sqrt{4 \sum_{\mathbf{a} \in \mathcal{K}_h(s)} \frac{\pi_h'(\mathbf{a}|s)^2}{n_h(s, \mathbf{a})} \log(4m \sum_{s \in \mathcal{S}, h \in [H]} \prod_{j \in [m]} |\mathcal{C}(\Pi_{h,j}(s))|(1 + HS/\epsilon_{\mathrm{cover}})^S/\delta)}$$

$$\leq H \sqrt{8 \sum_{\mathbf{a} \in \mathcal{K}_h(s)} \frac{\pi_h'(\mathbf{a}|s)^2}{n_h(s, \mathbf{a})} S \log(8m\mathcal{N}(\Pi)SH/\epsilon_{\mathrm{cover}}\delta)}.$$

Note that $r_{h,j}(s, \mathbf{a}) + \left\langle P_h(s, \mathbf{a}), \underline{V}_{h+1,j}^\pi \right\rangle - \widehat{r}_{h,j}(s, \mathbf{a}) - \left\langle \widehat{P}_h(s, \mathbf{a}), \underline{V}_{h+1,j}^\pi \right\rangle$ is bounded in $[-H, H]$ as $r_{h,j}(s, \mathbf{a}) \in [0, 1]$ and $\underline{V}_{h+1,j}^\pi \in [0, H - h]$. There exists $V_{h+1} \in \mathcal{V}$ such that $\|\underline{V}_{h+1,j}^\pi - V_{h+1}\|_\infty \leq \epsilon_{\mathrm{cover}}$, which implies

$$\left| \sum_{\mathbf{a} \in \mathcal{K}_h(s)} \pi_h'(\mathbf{a}|s) \left( r_h(s, \mathbf{a}) + \langle P_h(s, \mathbf{a}), V_{h+1} \rangle - \widehat{r}_h(s, \mathbf{a}) - \left\langle \widehat{P}_h(s, \mathbf{a}), V_{h+1} \right\rangle \right) \right|$$

$$- \left| \sum_{\mathbf{a} \in \mathcal{K}_h(s)} \pi_h'(\mathbf{a}|s) \left( r_h(s, \mathbf{a}) + \left\langle P_h(s, \mathbf{a}), \underline{V}_{h+1,j}^\pi \right\rangle - \widehat{r}_h(s, \mathbf{a}) - \left\langle \widehat{P}_h(s, \mathbf{a}), \underline{V}_{h+1,j}^\pi \right\rangle \right) \right|$$

$$\leq 2\epsilon_{\mathrm{cover}}.$$

For any $\pi_{h,j}^s \in \Pi_{h,j}(s)$, there exists $\pi_{h,j}'^s \in \mathcal{C}(\Pi_{h,j}(s))$ such that $\|\pi_{h,j}(\cdot|s) - \pi_{h,j}'(\cdot|s)\|_1 \leq \epsilon_{\mathrm{cover}}$ for all $j \in [m]$ and $s \in \mathcal{S}$. So with Lemma 29, we have

$$\left| \sum_{\mathbf{a} \in \mathcal{K}_h(s)} \pi_h'(\mathbf{a}|s) \left( r_{h,j}(s, \mathbf{a}) + \left\langle P_h(s, \mathbf{a}), \underline{V}_{h+1,j}^\pi \right\rangle - \widehat{r}_{h,j}(s, \mathbf{a}) - \left\langle \widehat{P}_h(s, \mathbf{a}), \underline{V}_{h+1,j}^\pi \right\rangle \right) \right.$$

$$- \sum_{\mathbf{a} \in \mathcal{K}_h(s)} \pi_h(\mathbf{a}|s) \left( r_h(s, \mathbf{a}) + \left\langle P_h(s, \mathbf{a}), \underline{V}^{\pi}_{h+1,j} \right\rangle - \widehat{r}_{h,j}(s, \mathbf{a}) - \left\langle \widehat{P}_h(s, \mathbf{a}), \underline{V}^{\pi}_{h+1,j} \right\rangle \right) \Bigg|$$

$$\leq m \epsilon_{\text{cover}} H.$$

By Lemma 30, we have

$$\left| \sqrt{\sum_{\mathbf{a} \in \mathcal{K}_h(s)} \frac{\pi'_h(\mathbf{a}|s)^2}{n_h(s, \mathbf{a})}} - \sqrt{\sum_{\mathbf{a} \in \mathcal{K}_h(s)} \frac{\pi_h(\mathbf{a}|s)^2}{n_h(s, \mathbf{a})}} \right| \leq \sqrt{2m\epsilon_{\text{cover}}}.$$

Combining all these parts together and then with probability $1 - \delta$, we have

$$\left| \sum_{\mathbf{a} \in \mathcal{K}_h(s)} \pi_h(\mathbf{a}|s) \left( r_{h,j}(s, \mathbf{a}) + \left\langle P_h(s, \mathbf{a}), \underline{V}^{\pi}_{h+1,j} \right\rangle - \widehat{r}_{h,j}(s, \mathbf{a}) - \left\langle \widehat{P}_h(s, \mathbf{a}), \underline{V}^{\pi}_{h+1,j} \right\rangle \right) \right|$$

$$\leq H \sqrt{8 \sum_{\mathbf{a} \in \mathcal{K}_h(s)} \frac{\pi_h(\mathbf{a}|s)^2}{n_h(s, \mathbf{a})} S \log(8m\mathcal{N}(\Pi, \epsilon_{\text{cover}})SH\delta)} + 2\epsilon_{\text{cover}} + m\epsilon_{\text{cover}}H$$

$$+ H \sqrt{8m\epsilon_{\text{cover}} \log(8m\mathcal{N}(\Pi, \epsilon_{\text{cover}})SH/\delta)}.$$

By Lemma 28, we have

$$\mathcal{N}(\Pi, \epsilon_{\text{cover}}) = \frac{1}{SH} \sum_{s \in \mathcal{S}, h \in [H]} \prod_{j \in [m]} |\mathcal{C}(\Pi_{h,j}(s), \epsilon_{\text{cover}})|$$

$$\leq \prod_{j \in [m]} (3A_j/\epsilon_{\text{cover}})^{A_j}$$

$$\leq (3(\sum_{j \in [m]} A_j)/\epsilon_{\text{cover}})^{\sum_{j \in [m]} A_j}.$$

Set $\epsilon_{\text{cover}} = \frac{1}{\sum_{j \in [m]} A_j m H^2 n^2}$ and with some calculations we can get

$$\left| \sum_{\mathbf{a} \in \mathcal{K}_h(s)} \pi_h(\mathbf{a}|s) \left( r_h(s, \mathbf{a}) + \left\langle P_h(s, \mathbf{a}), \underline{V}^{\pi}_{h+1,j} \right\rangle - \widehat{r}_{h,j}(s, \mathbf{a}) - \left\langle \widehat{P}_h(s, \mathbf{a}), \underline{V}^{\pi}_{h+1,j} \right\rangle \right) \right|$$

$$\leq H \sqrt{8 \sum_{\mathbf{a} \in \mathcal{K}_h(s)} \frac{\pi_h(\mathbf{a}|s)^2}{n_h(s, \mathbf{a})} S \log(8m\mathcal{N}(\Pi)SHn/\delta)} + \sqrt{32 \log(16 \prod_{j \in [m]} A_j m SHn/\delta)/n}$$

$$\leq H \sqrt{\sum_{\mathbf{a} \in \mathcal{K}_h(s)} \frac{\pi_h(a|s)^2}{n_h(s, \mathbf{a})} S \log(\mathcal{N}(\Pi))\iota} + \sqrt{\iota}/n.$$

$\square$

**Lemma 20.** *Under event $\mathcal{G}_{\text{marl}}$, for all $j \in [m]$, $h \in [H]$, $\pi \in \Pi$ and $s \in \mathcal{S}$, we have*

$$\underline{V}^{\pi}_{h,j}(s) \leq V^{\pi}_{h,j}(s) \leq \overline{V}^{\pi}_{h,j}(s).$$

*Proof.* We prove this argument by induction. It holds for $h = H + 1$ as $\underline{V}^{\pi}_{H+1,j}(s) = V^{\pi}_{H+1,j}(s) = \overline{V}^{\pi}_{H+1,j}(s)$. Suppose the argument holds for $h + 1$ and we consider $h$.

$$\underline{V}^{\pi}_{h,j}(s) = \text{proj}_{[0,H-h+1]} \left\{ \mathbb{E}_{\mathbf{a} \sim \pi_h(\cdot|s)} \widehat{r}_{h,j}(s, \mathbf{a}) + \mathbb{E}_{\mathbf{a} \sim \pi_h(\cdot|s)} \widehat{P}_h(s, \mathbf{a}) \cdot \underline{V}^{\pi}_{h+1,j} - b_h(s, \pi^s_h) \right\}$$

$$= \text{proj}_{[0,H-h+1]} \left\{ \sum_{\mathbf{a} \in \mathcal{K}_h(s)} \pi_h(\mathbf{a}|s) \left( \widehat{r}_{h,j}(s, \mathbf{a}) + \left\langle \widehat{P}_h(s, \mathbf{a}), \underline{V}^{\pi}_{h+1,j} \right\rangle \right) - b_h(s, \pi^s_h) \right\}$$

$$\leq \mathrm{proj}_{[0,H-h+1]} \left\{ \sum_{\mathbf{a}\in\mathcal{K}_h(s)} \pi_h(\mathbf{a}|s)\left(r_{h,j}(s,\mathbf{a}) + \left\langle P_h(s,\mathbf{a}), \underline{V}_{h+1,j}^\pi \right\rangle\right) \right\} \qquad \text{(Lemma 19)}$$

$$\leq \mathrm{proj}_{[0,H-h+1]} \left\{ \sum_{\mathbf{a}\in\mathcal{K}_h(s)} \pi_h(\mathbf{a}|s)\left(r_{h,j}(s,\mathbf{a}) + \left\langle P_h(s,\mathbf{a}), V_{h+1,j}^\pi \right\rangle\right) \right\}$$
$$\text{(Induction hypothesis)}$$

$$\leq \mathrm{proj}_{[0,H-h+1]} \left\{ \sum_{\mathbf{a}\in\mathcal{A}} \pi_h(\mathbf{a}|s)\left(r_{h,j}(s,\mathbf{a}) + \left\langle P_h(s,\mathbf{a}), V_{h+1,j}^\pi \right\rangle\right) \right\}$$

$$\leq \mathrm{proj}_{[0,H-h+1]} \left\{ V_{h,j}^\pi(s) \right\}$$
$$= V_{h,j}^\pi(s).$$

$$\overline{V}_{h,j}^\pi(s)$$

$$= \mathrm{proj}_{[0,H-h+1]} \left\{ \mathbb{E}_{\mathbf{a}\sim\pi_h(\cdot|s)}\widehat{r}_{h,j}(s,\mathbf{a}) + \mathbb{E}_{\mathbf{a}\sim\pi_h(\cdot|s)}\widehat{P}_h(s,\mathbf{a})\cdot\overline{V}_{h+1,j}^\pi + b_h(s,\pi_h^s) + H\sum_{a\notin\mathcal{K}(s)}\pi_h(\mathbf{a}|s) \right\}$$

$$= \mathrm{proj}_{[0,H-h+1]} \left\{ \sum_{\mathbf{a}\in\mathcal{K}_h(s)}\pi_h(\mathbf{a}|s)\left(\widehat{r}_{h,j}(s,\mathbf{a}) + \left\langle\widehat{P}_h(s,\mathbf{a}),\overline{V}_{h+1,j}^\pi\right\rangle\right) + b_h(s,\pi_h^s) + H\sum_{a\notin\mathcal{K}(s)}\pi_h(\mathbf{a}|s) \right\}$$

$$\geq \mathrm{proj}_{[0,H-h+1]} \left\{ \sum_{\mathbf{a}\in\mathcal{K}_h(s)}\pi_h(\mathbf{a}|s)\left(r_{h,j}(s,\mathbf{a}) + \left\langle P_h(s,\mathbf{a}),\overline{V}_{h+1,j}^\pi\right\rangle\right) + H\sum_{a\notin\mathcal{K}(s)}\pi_h(\mathbf{a}|s) \right\}$$
$$\text{(Lemma 19)}$$

$$\geq \mathrm{proj}_{[0,H-h+1]} \left\{ \sum_{\mathbf{a}\in\mathcal{A}}\pi_h(\mathbf{a}|s)\left(r_{h,j}(s,\mathbf{a}) + \left\langle P_h(s,\mathbf{a}),\overline{V}_{h+1,j}^\pi\right\rangle\right) \right\}$$
$$(\overline{V}_{h+1,j}^\pi(s)\leq H-h \text{ for all } s\in\mathcal{S})$$

$$\geq \mathrm{proj}_{[0,H-h+1]} \left\{ \sum_{\mathbf{a}\in\mathcal{A}}\pi_h(\mathbf{a}|s)\left(r_{h,j}(s,\mathbf{a}) + \left\langle P_h(s,\mathbf{a}),V_{h+1,j}^\pi\right\rangle\right) \right\} \qquad \text{(Induction hypothesis)}$$

$$= \mathrm{proj}_{[0,H-h+1]} \left\{ V_{h,j}^\pi(s) \right\}$$
$$= V_{h,j}^\pi(s).$$

$\square$

**Lemma 21.** *Under event $\mathcal{G}_{\mathrm{marl}}$, for any policy $\pi\in\Pi$, we have*
$$\mathrm{Gap}(\pi) \leq \sum_{j\in[m]} \overline{V}_{1,j}^{*,\pi^{-j}}(s) - \underline{V}_{1,j}^\pi(s).$$

*In addition, we have*
$$\mathrm{Gap}(\pi^{\mathrm{output}}) \leq \min_{\pi\in\Pi}\sum_{j\in[m]}\left[\overline{V}_{1,j}^{*,\pi^{-j}}(s) - \underline{V}_{1,j}^\pi(s)\right].$$

*Proof.* By Lemma 20, we have
$$\mathrm{Gap}(\pi) = \max_{\pi'}\sum_{j\in[m]} V_{1,j}^{\pi_j',\pi^{-j}}(s) - V_{1,j}^\pi(s) \leq \max_{\pi'}\sum_{j\in[m]}\overline{V}_{1,j}^{\pi_j',\pi^{-j}}(s) - \underline{V}_{1,j}^\pi(s).$$

Combined with Lemma 17 we can prove the first argument. For the second argument, note that $\pi_{\mathrm{output}}$ is the minimizer of the RHS, so we have
$$\mathrm{Gap}(\pi^{\mathrm{output}}) \leq \min_{\pi\in\Pi}\sum_{j\in[m]}\overline{V}_{1,j}^{*,\pi^{-j}}(s) - \underline{V}_{1,j}^\pi(s).$$

$\square$

**Lemma 22.** *Under event* $\mathcal{G}_{\mathrm{marl}}$, *for any strategy* $\pi \in \Pi$, *we have*

$$\underline{V}^{\pi}_{1,j}(s_1) \geq V^{\pi}_{1,j}(s_1) - \mathbb{E}_{\pi} \sum_{h \in [H]} \widehat{b}_h(s_h, \pi_h^{s_h}), \ \overline{V}^{\pi}_{1,j}(s_1) \leq V^{\pi}_{1,j}(s_1) + \mathbb{E}_{\pi} \sum_{h \in [H]} \widehat{b}_h(s_h, \pi_h^{s_h}).$$

*Proof.* We prove the first argument and the second argument holds similarly.

$$V^{\pi}_{1,j}(s_1) - \underline{V}^{\pi}_{1,j}(s_1)$$

$$= \mathbb{E}_{\mathbf{a} \sim \pi_1(\cdot|s_1)} \left[ r_{1,j}(s_1, \mathbf{a}) + P_1(s_1, \mathbf{a}) \cdot V^{\pi}_{2,j} \right] - \mathbb{E}_{\mathbf{a} \sim \pi_1(\cdot|s_1)} \left[ \widehat{r}_{1,j}(s_1, \mathbf{a}) + \widehat{P}_1(s_1, \mathbf{a}) \cdot \underline{V}^{\pi}_{2,j} \right] + b_1(s_1, \pi_1^{s_1})$$

$$= \mathbb{E}_{\pi_1} \left[ V^{\pi}_{2,j}(s_2) - \underline{V}^{\pi}_{2,j}(s_2) \right] + \mathbb{E}_{\pi_1} \left[ r_{1,j}(s_1, \mathbf{a}) + P_1(s_1, \mathbf{a}) \cdot \underline{V}^{\pi}_{2,j} - \widehat{r}_{1,j}(s_1, \mathbf{a}) - \widehat{P}_1(s_1, \mathbf{a}) \cdot \underline{V}^{\pi}_{2,j} \right] + b_1(s_1, \pi_1^{s_1})$$

$$\leq \mathbb{E}_{\pi_1} \left[ V^{\pi}_{2,j}(s_2) - \underline{V}^{\pi}_{2,j}(s_2) \right] + \sum_{\mathbf{a} \in \mathcal{K}_h(s_1)} \pi_1(\mathbf{a}|s_1) \left( r_{1,j}(s_1, \mathbf{a}) + P_1(s_1, \mathbf{a}) \cdot \underline{V}^{\pi}_{2,j} - \widehat{r}_{1,j}(s_1, \mathbf{a}) - \widehat{P}_1(s_1, \mathbf{a}) \cdot \underline{V}^{\pi}_{2,j} \right)$$

$$+ \sum_{\mathbf{a} \notin \mathcal{K}_h(s_1)} \pi(\mathbf{a}|s_1) H + b_1(s_1, \pi_1^{s_1})$$

$$\leq \mathbb{E}_{\pi_1} \left[ V^{\pi}_{2,j}(s_2) - \underline{V}^{\pi}_{2,j}(s_2) \right] + \sum_{\mathbf{a} \notin \mathcal{K}_h(s_1)} \pi_1(\mathbf{a}|s_1) H + 2 b_1(s_1, \pi_1^{s_1})$$

$$= \mathbb{E}_{\pi_1} \left[ V^{\pi}_{2,j}(s_2) - \underline{V}^{\pi}_{2,j}(s_2) \right] + \widehat{b}_1(s_1, \pi_1^{s_1}).$$

By telescoping we can prove the first argument. $\qquad\square$

**Lemma 23.** *Under good event* $\mathcal{G}_{\mathrm{marl}}$, *for any strategy* $\pi \in \Pi$, *we have*

$$\sum_{j \in [m]} \overline{V}^{*,\pi_{-j}}_{1,j}(s_1) - \underline{V}^{\pi}_{1,j}(s_1) \leq \mathrm{Gap}(\pi) + \max_{\pi' \in \Pi^{\mathrm{det}}} \sum_{j \in [m]} \mathbb{E}_{\pi'_j, \pi_{-j}} \left[ \sum_{h=1}^{H} \widehat{b}_h(s_h, \pi'^{s_h}_{h,j}, \pi^{s_h}_{h,-j}) \right] + m \mathbb{E}_{\pi} \sum_{h=1}^{H} \left[ \widehat{b}_h(s_h, \pi^{s_h}_h) \right].$$

*Proof.* Set $\widetilde{\pi} = \mathrm{argmax}_{\pi' \in \Pi^{\mathrm{full}}} \sum_{j \in [m]} \overline{V}^{\pi'_j, \pi_{-j}}_{1,j}(s_1) - \underline{V}^{\pi}_{1,j}(s_1)$. Lemma 17 shows that there always exists a deterministic strategy $\widetilde{\pi} \in \Pi^{\mathrm{det}}$, which is used by Algorithm 3.

$$\max_{\pi' \in \Pi^{\mathrm{full}}} \sum_{j \in [m]} \overline{V}^{\pi'_j, \pi_{-j}}_{1,j}(s_1) - \underline{V}^{\pi}_{1,j}(s_1)$$

$$= \sum_{j \in [m]} \overline{V}^{\widetilde{\pi}_j, \pi_{-j}}_{1,j}(s_1) - \underline{V}^{\pi}_{1,j}(s_1)$$

$$\leq \sum_{j \in [m]} \left[ V^{\widetilde{\pi}_j, \pi_{-j}}_{1,j}(s_1) - V^{\pi}_{1,j}(s_1) + \mathbb{E}_{\widetilde{\pi}_j, \pi_{-j}} \sum_{h \in [H]} \widehat{b}_h(s_h, \widetilde{\pi}^{s_h}_{h,j}, \pi^{s_h}_{h,-j}) + \mathbb{E}_{\pi} \sum_{h \in [H]} \widehat{b}_h(s_h, \pi^{s_h}_h) \right]$$

$$\text{(Lemma 22)}$$

$$\leq \max_{\pi' \in \Pi^{\mathrm{det}}} \sum_{j \in [m]} \left[ V^{\pi'_j, \pi_{-j}}_{1,j}(s_1) - V^{\pi}_{1,j}(s_1) \right] + \sum_{j \in [m]} \mathbb{E}_{\widetilde{\pi}_j, \pi_{-j}} \left[ \sum_{h=1}^{H} \widehat{b}_h(s_h, \widetilde{\pi}^{s_h}_{h,j}, \pi^{s_h}_{h,-j}) \right] + m \mathbb{E}_{\pi} \sum_{h=1}^{H} \left[ \widehat{b}_h(s_h, \pi^{s_h}_h) \right]$$

$$\leq \mathrm{Gap}(\pi) + \max_{\pi' \in \Pi^{\mathrm{det}}} \sum_{j \in [m]} \mathbb{E}_{\pi'_j, \pi_{-j}} \left[ \sum_{h=1}^{H} \widehat{b}_h(s_h, \pi'^{s_h}_{h,j}, \pi^{s_h}_{h,-j}) \right] + m \mathbb{E}_{\pi} \sum_{h=1}^{H} \left[ \widehat{b}_h(s_h, \pi^{s_h}_h) \right].$$

$$\qquad\square$$

### D.1 Dataset-dependent Bound

**Lemma 24.** *Suppose* $\widehat{C}(\pi)$ *is finite. For any strategy* $\pi' \in \Pi$, $h \in [H]$ *and* $j \in [m]$, *we have*

$$\mathbb{E}_{\pi'_j, \pi_{-j}} b_h(s_h, \pi'^{s_h}_{h,j}, \pi^{s_h}_{h,-j}) \leq 2HS \sqrt{\widehat{C}(\pi) \log(\mathcal{N}(\Pi)) \iota / n}.$$

*Proof.*

$$\mathbb{E}_{\pi'_j, \pi_{-j}} b_h(s_h, \pi'^{s_h}_{h,j}, \pi^{s_h}_{h,-j})$$

$$=\mathbb{E}_{\pi'_j, \pi_{-j}} H \sqrt{\sum_{\mathbf{a} \in \mathcal{K}_h(s_h)} \frac{(\pi'_{h,j}, \pi_{h,-j})(\mathbf{a}|s_h)^2}{n_h(s, \mathbf{a})} S \log(\mathcal{N}(\Pi)) \iota + \sqrt{\iota}/n}$$

$$=\sum_{s_h \in \mathcal{S}} H \sqrt{\sum_{\mathbf{a} \in \mathcal{K}_h(s_h)} \frac{d_h^{\pi'_j, \pi_{-j}}(s_h)(\pi'_{h,j}, \pi_{h,-j})(\mathbf{a}|s_h)^2}{n_h(s_h, \mathbf{a})} S \log(\mathcal{N}(\Pi)) \iota + \sqrt{\iota}/n}$$

$$=\sum_{s_h \in \mathcal{S}} H \sqrt{\sum_{\mathbf{a} \in \mathcal{K}_h(s_h)} \frac{d_h^{\pi'_j, \pi_{-j}}(s_h, \mathbf{a})^2}{n\widehat{d}_h(s_h, \mathbf{a})} S \log(\mathcal{N}(\Pi)) \iota + \sqrt{\iota}/n}$$

$$\leq \sum_{s_h \in \mathcal{S}} H \sqrt{\sum_{\mathbf{a} \in \mathcal{K}_h(s_h)} \widehat{C}(\pi) d_h^{\pi'_j, \pi_{-j}}(s_h, \mathbf{a}) S \log(\mathcal{N}(\Pi)) \iota/n + \sqrt{\iota}/n}$$

$$\leq H \sqrt{S^2 \widehat{C}(\pi) \log(\mathcal{N}(\Pi)) \iota/n + \sqrt{\iota}/n} \qquad \text{(Cauchy-Schwarz inequality)}$$

$$\leq 2HS \sqrt{\widehat{C}(\pi) \log(\mathcal{N}(\Pi)) \iota/n}.$$

$\square$

**Lemma 25.** *Suppose $\widehat{C}(\pi)$ is finite. For any strategy $\pi' \in \Pi$, $h \in [H]$ and $j \in [m]$, we have*

$$\mathbb{E}_{\pi'_j, \pi_{-j}} \sum_{\mathbf{a}_h \notin \mathcal{K}_h(s_h)} (\pi'_{h,j}, \pi_{h,-j})(\mathbf{a}_h|s_h) = 0.$$

*Proof.* Similar to Lemma 11, we have

$$\mathbb{E}_{\pi'_j, \pi_{-j}} \sum_{\mathbf{a}_h \notin \mathcal{K}_h(s_h)} (\pi'_{h,j}, \pi_{h,-j})(\mathbf{a}_h|s_h)$$

$$=\mathbb{E}_{\pi'_j, \pi_{-j}} \sum_{\mathbf{a}_h : \widehat{d}_h(s_h, \mathbf{a}_h)=0} (\pi'_{h,j}, \pi_{h,-j})(\mathbf{a}_h|s_h)$$

$$= \sum_{\mathbf{a} : \widehat{d}_h(s_h, \mathbf{a}_h)=0} d_h^{\pi'_j, \pi_{-j}}(s_h, \mathbf{a}_h)$$

$$\leq \widehat{C}(\pi) \sum_{\mathbf{a} : \widehat{d}_h(s_h, \mathbf{a}_h)=0} \widehat{d}_h(s_h, \mathbf{a}_h)$$

$$=0.$$

$\square$

**Lemma 26.** *For any strategy $\pi \in \Pi$ and $j \in [m]$, we have*

$$\max_{\pi'} \mathbb{E}_{\pi'_j, \pi_{-j}} \left[ \sum_{h=1}^H \widehat{b}_h(s_h, \pi'^{s_h}_{h,j}, \pi^{s_h}_{h,-j}) \right] \leq 2H^2 S \sqrt{\widehat{C}(\pi) \log(\mathcal{N}(\Pi)) \iota/n}.$$

*Proof.* If $\widehat{C}(\pi)$ is infinite, the argument holds directly. Otherwise it can be derived from Lemma 24 and Lemma 25. $\square$

**Theorem 7.** *With probability $1 - \delta$, we have*

$$\mathrm{Gap}(\pi^{\mathrm{output}}) \leq \min_{\pi \in \Pi} \left[ \mathrm{Gap}(\pi) + 4mH^2 S \sqrt{\widehat{C}(\pi) \log(\mathcal{N}(\Pi)) \iota/n} \right].$$

*Proof.* This can be derived from Lemma 26, Lemma 21 and Lemma 23. $\square$

## D.2 Dataset-independent Bound

**Lemma 27.** *Suppose* $p_{\min} = \min_{s,\mathbf{a},h}\{d_h^\rho(s,\mathbf{a}) : d_h^\rho(s,\mathbf{a}) > 0\}$. *With probability* $1 - \delta$, *for all* $h, s, \mathbf{a}$, *we have*

$$n_h(s,\mathbf{a}) \geq \left(1 - \sqrt{\frac{2\log(S\Pi_{j\in[m]}A_j H/\delta)}{np_{\min}}}\right) nd_h(s,\mathbf{a}).$$

*As a result, if* $n \geq \frac{8\log(S\Pi_{j\in[m]}A_j H/\delta)}{p_{\min}}$, *for all strategy* $\pi$, *we have*

$$2C(\pi) \geq \widehat{C}(\pi).$$

*Proof.* For a fixed $s, \mathbf{a}, h$, for any $\epsilon > 0$ we have

$$\mathbb{P}(n_h(s,\mathbf{a}) < (1-\epsilon)nd_h(s,\mathbf{a})) \leq \exp\left(-\frac{\epsilon^2 nd_h(s,\mathbf{a})}{2}\right) \leq \exp\left(-\frac{\epsilon^2 np_{\min}}{2}\right).$$

With a union bound, we have

$$\mathbb{P}(\exists h, s, a, b : \mathbb{P}(n_h(s,a,b) < (1-\epsilon)nd_h(s,a,b))) \leq S\Pi_{j\in[m]}A_j H \exp\left(-\frac{\epsilon^2 np_{\min}}{2}\right).$$

The RHS is smaller than $\delta$ if we set

$$\epsilon = \sqrt{\frac{2\log(S\Pi_{j\in[m]}A_j H/\delta)}{np_{\min}}}.$$

If $n \geq \frac{8\log(S\Pi_{j\in[m]}A_j H/\delta)}{p_{\min}}$, we have $\widehat{d}_h(s,\mathbf{a}) = \frac{n_h(s,\mathbf{a})}{n} \geq \frac{d_h(s,\mathbf{a})}{2}$. By Definition 3 and Definition 2, we have

$$2C(\pi) \geq \widehat{C}(\pi).$$

$\square$

**Theorem 8.** *If* $n \geq \frac{8\log(S\Pi_{j\in[m]}A_j H/\delta)}{p_{\min}}$, *with probability* $1 - \delta$, *we have*

$$\mathrm{Gap}(\pi^{\mathrm{output}}) \leq \min_{\pi\in\Pi}\left[\mathrm{Gap}(\pi) + 4mH^2 S\sqrt{2C(\pi)\log(\mathcal{N}(\Pi)\iota/n)}\right].$$

*Proof.* This can be derived by Lemma 27 and Theorem 7. $\square$

# E Technical Lemmas

**Lemma 28.** *(L-1 covering number of probability simplex) For probability simplex* $\Delta(\mathcal{A})$ *and* $A = |\mathcal{A}|$, *there exists a subset* $\Delta'(\mathcal{A}) \subset \Delta(\mathcal{A})$ *such that for any* $p \in \Delta(\mathcal{A})$, *there exists* $p' \in (\mathcal{A})$ *such that* $\|p - p'\|_1 \leq \epsilon$. *In addition,*

$$|\Delta'(\mathcal{A})| \leq \left(\frac{3A}{\epsilon}\right)^A.$$

*Proof.* We construct $\epsilon'$-net for $\epsilon/2 < \epsilon' \leq \epsilon$ such that $1/\epsilon'$ is integer. Then this $\epsilon'$-net is directly a $\epsilon$-net as $\epsilon' \leq \epsilon$. Define $D(\mathcal{A}) = \{(n_1\epsilon', n_2\epsilon', \cdots, n_A\epsilon'), \sum_{i=1}^A = \frac{1}{\epsilon'}, n_i \in [0, 1/\epsilon']\} \subset \Delta(\mathcal{A})$. For $p = (p_1, p_2, \cdots, p_A) \in \Delta(\mathcal{A})$, suppose

$$k_i\epsilon' \leq p_i < (k_i+1)\epsilon',$$

for some non-negative integers $\{k_i\}$. Set $k = \sum_{i=1}^A k_i$ Then we have $1/\epsilon' - A < k \leq 1/\epsilon'$. Now we construct $p' = (n_1\epsilon', n_2\epsilon', \cdots, n_A\epsilon') \in D(\mathcal{A})$ such that

$$\begin{cases} n_i = k_i + 1, & i \in [1/\epsilon' - k] \\ n_i = k_i, & \text{otherwise.} \end{cases}$$

Then we have $|p_i - p_i'| \leq \epsilon'$ for all $i \in [A]$, which implies

$$\|p - p'\| \leq A\epsilon'.$$

So $|D(\mathcal{A})| \leq \left(\frac{1+\epsilon'}{\epsilon'}\right)^A \leq \left(\frac{3}{\epsilon}\right)^A$ is an $A\epsilon$-net of $\Delta(\mathcal{A})$. We can prove the lemma by rescaling $\epsilon$. $\square$

**Lemma 29.** *Suppose $\pi_j, \pi_j' \in \Delta(\mathcal{A}_j)$ such that $\left\|\pi_j - \pi_j'\right\|_1 \leq \epsilon$ for all $j \in [m]$. For any function $f(\mathbf{a}) \in [-H, H]$, we have*

$$|\mathbb{E}_{\mathbf{a} \sim \pi} f(\mathbf{a}) - \mathbb{E}_{\mathbf{a} \sim \pi'} f(\mathbf{a})| \leq m\epsilon H.$$

*Proof.*

$$|\mathbb{E}_{\mathbf{a} \sim \pi} f(\mathbf{a}) - \mathbb{E}_{\mathbf{a} \sim \pi'} f(\mathbf{a})|$$

$$= \left| \sum_{\mathbf{a}} \Pi_{j=1}^m \pi_j(a_j) f(\mathbf{a}) - \sum_{\mathbf{a}} \Pi_{j=1}^m \pi_j'(a_j) f(\mathbf{a}) \right|$$

$$= \left| \sum_{j=1}^m \sum_{\mathbf{a}_{-j} \in \Pi_{i \neq j} \mathcal{A}_i} \Pi_{i=1}^{j-1} \pi_i(a_i) \Pi_{i=j+1}^m \pi_i'(a_i) \sum_{a_j \in \mathcal{A}_j} \left( \pi_j(a_j) - \pi_j'(a_j) \right) f(\mathbf{a}) \right|$$

$$\leq \left| \sum_{j=1}^m \sum_{\mathbf{a}_{-j} \in \Pi_{i \neq j} \mathcal{A}_i} \Pi_{i=1}^{j-1} \pi_i(a_i) \Pi_{i=j+1}^m \pi_i'(a_i) \epsilon H \right|$$

$$= m\epsilon H.$$

$\square$

**Lemma 30.** *Suppose $\pi_j, \pi_j' \in \Delta(\mathcal{A}_j)$ such that $\left\|\pi_j - \pi_j'\right\|_1 \leq \epsilon$ for all $j \in [m]$. For any set $\mathcal{K} \subset \Pi_{j \in [m]} \mathcal{A}_j$ and function $n(\mathbf{a}) \geq 1$ we have*

$$\left| \sqrt{\sum_{\mathbf{a} \in \mathcal{K}} \frac{\pi(\mathbf{a})^2}{n(\mathbf{a})}} - \sqrt{\sum_{\mathbf{a} \in \mathcal{K}} \frac{\pi'(\mathbf{a})^2}{n(\mathbf{a})}} \right| \leq \sqrt{2m\epsilon}.$$

*Proof.*

$$\left| \sqrt{\sum_{\mathbf{a} \in \mathcal{K}} \frac{\pi(\mathbf{a})^2}{n(\mathbf{a})}} - \sqrt{\sum_{\mathbf{a} \in \mathcal{K}} \frac{\pi'(\mathbf{a})^2}{n(\mathbf{a})}} \right|$$

$$\leq \sqrt{\left| \sum_{\mathbf{a} \in \mathcal{K}} \frac{\pi(\mathbf{a})^2 - \pi'(\mathbf{a})^2}{n(\mathbf{a})} \right|}$$

$$= \sqrt{\left| \sum_{j=1}^m \sum_{\mathbf{a}_{-j} \in \prod_{i \neq j} \mathcal{A}_i} \prod_{i=1}^{j-1} \pi_i^2(a_i) \prod_{i=j+1}^m \pi_i'^2(a_i) \sum_{a_j \in \mathcal{A}_j} \left( \pi_j^2(a_j) - \pi_j'^2(a_j) \right) \mathbf{1}(\mathbf{a} \in \mathcal{K})/n(\mathbf{a}) \right|}$$

$$\leq \sqrt{\left| \sum_{j=1}^m \sum_{\mathbf{a}_{-j} \in \prod_{i \neq j} \mathcal{A}_i} \prod_{i=1}^{j-1} \pi_i^2(a_i) \prod_{i=j+1}^m \pi_i'^2(a_i) 2\epsilon \right|}$$

$$\leq \sqrt{2m\epsilon}.$$

$\square$