# OpenReview forum: "Provably Efficient Offline Multi-agent Reinforcement Learning via Strategy-wise Bonus"
_NeurIPS.cc/2022/Conference — NeurIPS 2022 Accept_

### Official Review · Reviewer_5fGy · 2022-07-11

**Rating:** 4
**Confidence:** 5
**Soundness:** 1 poor
**Presentation:** 2 fair
**Contribution:** 2 fair

**Summary:**

This paper considers offline multi-agent reinforcement learning.

**Questions:**

The sample complexity for multi-agent general sum MDP is exponential.

**Limitations:**

The sample complexity for multi-agent general sum MDP is exponential.

**Strengths And Weaknesses:**

The sample complexity for multi-agent general sum MDP is at least $N = \prod_{i=1}^m A_i$, where $A_i$'s denote the effective action space size for each agent (i.e., the number of actions obeying $\pi_i(a) > 0$).
Specifically, to ensure $\hat{C} < \infty$, one needs at least $N$ samples such that ${\hat{d}}h(s, a) > 0$ when $d_h^{\pi}(s, a) > 0$.
In addition, the assumption $C < \infty$ implies there are at least $N$ actions satisfying $d_h(s, a) > 0$, which leads to $1/p_{\min} \ge N$.
Hence, all results for the multi-agent case in Section 3.2 exhibit an exponential dependence on the number of agents, except the case that almost all agents will choose only one action.

On the other hand, it is well known to find the NE for multi-agent case is computationally hard. Then is it possible to compute CCE effectively for offline multi-agent RL?

---

> ### Author Response · Authors · 2022-07-31
> **Response to Reviewer 5fGy**
>
> Thanks for your careful reading! We address your concerns below.
>
> - **Exponential Dependence:** You are correct that in the **worst case** $n$ needs to be exponentially large to reach the provided guarantees as a burn-in cost. However, this does not deny our contributions.
>   - First, for the offline two-player zero-sum Markov games, Corollary 2 has no such burn-in cost and the resulting sample complexity perfectly improves the previous $AB$ dependence to $A+B$.
>   - Second, for offline general-sum Markov games, there is no existing algorithm before and it is even unknown if we can learn the NE with **infinite data** under the unilateral coverage assumption (note the necessity result of unilateral assumption holds for infinite data regime [Cui and Du, 2022])
>   - Third, using a point-wise bonus in our algorithm will make the main order term scales as $mH^2S\sqrt{\Pi_{j\in[m]}A_jC(\pi)/n}$ while our strategy-wise bonus reduce it to $mH^2S\sqrt{\sum_{j\in[m]}A_jC(\pi)/n}$, and this is why we say we remove the exponential dependence.
>   - Fourth, indeed the burn-in cost can be exponential, but this is a problem-dependent term. For example, as you have mentioned, if the Nash equilibrium has a polynomial-sized support on the action space (e.g., deterministic Nash equilibrium in Markov potential games/Markov cooperative games [1]), then the burn-in cost can also be polynomial.
> - **CCE in Offline MARL:** This is a good question! First, our method can be directly applied to finding CCE, but as the CCE does not have the product structure as NE, strategy-wise bonus can not be used to remove the exponential dependence. In addition, this algorithm is still not computationally efficient. Indeed, it is unclear if it is possible to find CCE computationally efficiently under the unilateral coverage assumption. On the other hand, if we have the uniform coverage, we believe computing the CCE directly with respect to the empirical Markov game may work.
>
> ```
> [1] Ding, Dongsheng, et al. "Independent policy gradient for large-scale markov potential games: Sharper rates, function approximation, and game-agnostic convergence." International Conference on Machine Learning. PMLR, 2022.
> ```

---

> ### Author Response · Authors · 2022-08-07
> **Any other questions to be addressed？**
>
> Thanks again for your review. We hope our answers could increase your confidence. As the discussion period is close to the end and we have not yet heard back from you, we would be glad to see if our rebuttal response has addressed your concerns questions/concerns.
> We are more than happy to discuss further if you have any further concerns and issues, please kindly let us know your feedback. Thank you for your time and help!

---

### Official Review · Reviewer_1W3v · 2022-07-14

**Rating:** 7
**Confidence:** 3
**Soundness:** 3 good
**Presentation:** 3 good
**Contribution:** 3 good

**Summary:**

In this paper, the authors study the offline multi-agent RL. They propose the strategy-wise concentration principle in contrast to the point-wise concentration principle in existing works. By the new concentration principle, they propose novel algorithms with strategy-wise bonuses for both zero-sum and general-sum games, which achieve better sample complexities than the existing works. The sample complexity of the new method does not scale with the size of the joint action space. Moreover, the proposed algorithms can take a pre-specified strategy class as input and output a strategy that is close to the best strategy in such a class.

**Questions:**

I have a small question regarding this strategy-wise concentration method. Is it possible to apply such a strategy-wise bonus to the online Markov game setting?

**Limitations:**

I was not able to find the discussion of the limitation. It would be much better if the authors can explicitly summarize the limitation of their work in the conclusion section or in the checklist.

**Strengths And Weaknesses:**

This paper proposes novel algorithms with the strategy-wise bonus, which leads to better sample complexities than the existing works. I think it is an important contribution to the theoretical analysis of the offline Markov games as the new sample complexity result does not scale with the size of the joint action space. In addition, the writing of this work is good in general and this paper is well structured.

However, I was not able to find the discussion of the limitation in the paper although the author answered yes in the checklist. It would be much better if the authors can explicitly summarize the limitation of their work in the conclusion section or in the checklist.

Besides, is it possible to apply such a strategy-wise bonus to the online Markov game setting?

---

> ### Author Response · Authors · 2022-07-31
> **Response to Reviewer 1W3v**
>
> Thanks for your appreciation! We address your concerns below.
>
> - **Limitations:** One limitation is the computational inefficiency for finding NE in offline multi-player general-sum games, which is generally unavoidable given the hardness result [Daskalakism, 2013]. It would be interesting to see if other kinds of equilibrium (i.e., correlated equilibrium or coarse correlated equilibrium) can be solved efficiently in the offline setting.
> - **Application to Online Markov Game:** We believe it is possible to apply strategy-wise bonus to online Markov games to derive a tighter bound. This may be a promising future direction.

---

> ### Author Response · Authors · 2022-08-07
> **Any other questions to be addressed？**
>
> Thanks again for your review. We hope our answers could increase your confidence. As the discussion period is close to the end and we have not yet heard back from you, we would be glad to see if our rebuttal response has addressed your concerns questions/concerns.
> We are more than happy to discuss further if you have any further concerns and issues, please kindly let us know your feedback. Thank you for your time and help!

---

### Official Review · Reviewer_ctbY · 2022-07-15

**Rating:** 6
**Confidence:** 3
**Soundness:** 3 good
**Presentation:** 3 good
**Contribution:** 2 fair

**Summary:**

While there is an extensive literature on both multi-agent reinforcement learning (MARL) and offline RL separately, this work explores a setting that combines the two. In the MARL setting, the size of the joint action space grows exponentially with the number of agents, which is a key challenge in this subarea. This paper investigates how to tackle this problem and whether we can find a Nash equilibrium strategy in the offline MARL setting given unilateral coverage.
Specifically, to construct a confidence interval for each state-action pair in the offline MARL setting, the authors proposes to estimate each strategy separately to circumvent the dependence of the joint action space which grows exponentially with the number of agents. Besides, based on this, the authors also develop two different algorithm frameworks under the tabular case for both the offline two-player zero-sum Markov games as well as the offline multi-player general-sum Markov games separately.



**Questions:**

The analysis of this paper is based on a pre-specified policy class that contains the NE. The structure of the strategy class is a critical component in the proposed algorithm frameworks, which is especially true for the efficient computation of the surrogate function in the multi-player general-sum Markov game setting. Is a principled approach for structuring the strategy class?
In addition, while the strategy-wise bonus can avoid the exponential dependence of the number of players, what the potential downsides or limitations of this approach compared to point-wise estimations?

**Limitations:**

The authors has addressed some potential limitations.

**Strengths And Weaknesses:**

This paper helps to improve our current understanding of algorithms for the offline MARL under the tabular setting. The major contribution is the proposed strategy-wise bonus, which firstly estimates the state value functions for a strategy and then add the bonus on them. However, if the strategies are complicated and therefore the class space not well structured, the calculation for state value functions for the strategy could be challenging.

Given this, for the two-player zero-sum Markov games, the proposed framework utilizes a strategy-wise bonus as well as maximin-optimization-type algorithm. For the multi-player general-sum Markov game, as there is no saddle point structure, this paper proposes a surrogate function to minimize the performance gap, which, however, requires to enumerate the strategy class in the worst case and thus not computationally efficient.

Another concern regarding the current version is that the data assumption adopted in this work (Assumption 1) assumes that the all the tuples in the logged dataset are independent, while this is not true in most real-world scenarios. As in many real-world applications, this assumption is very restrictive on the dataset and therefore usually does not hold. To derive practical algorithms based on the theoretical analysis and test the efficacy of the proposed methods in practice, the work should consider the impact of the violation of this assumption on the algorithm performance.

---

> ### Author Response · Authors · 2022-07-31
> **Response to Reviewer ctbY**
>
> Thanks for your appreciation! We address your concerns below.
>
> - **Computational Efficiency:** The computational inefficiency for finding NE in offline general-sum Markov games is generally unavoidable as computing NE is PPAD-hard even for full-information general-sum games [Daskalakism, 2013]. However, if the pre-specified strategy class is well-structured, it might be possible to compute NE efficiently and we leave this to future work.
> - **Data Independence Assumption:** In many settings, we collect trajectory data so that different trajectories are independent while the data in the same trajectory are correlated. In particular, as mentioned in Line 171, the standard subsampling technique (e.g., in Li et al. [2022]) can be used to create a dataset that satisfies our assumption. Other kinds of dependency in the dataset are beyond the scope of this paper and it is not well understood even for single-agent offline RL, so this may be an interesting future direction.
> - **Structure of Prespecified Policy Class:** The policy class is usually constructed by some prior knowledge. For instance, we may want to find a policy that has a small NE gap in a given finite policy class. In applications, researchers usually parameterize the policy class by neural networks, see Mnih et al. [2016], Lowe et al. [2017], Haarnoja et al. [2018].
> - **Downside of Strategy-wise bonus:** We are not aware of any downside of using strategy-wise bonus and we believe it can be applied to other MARL settings.

---

> ### Author Response · Authors · 2022-08-07
> **Any other questions to be addressed？**
>
> Thanks again for your review. We hope our answers could increase your confidence. As the discussion period is close to the end and we have not yet heard back from you, we would be glad to see if our rebuttal response has addressed your concerns questions/concerns.
> We are more than happy to discuss further if you have any further concerns and issues, please kindly let us know your feedback. Thank you for your time and help!

---

### Meta-Review · Area_Chair_wyhG · 2022-08-27

**Recommendation:** Accept
**Confidence:** Certain

**Metareview:**

Reviewers appreciate the paper's contribution to a novel intersection of fields: offline and multi-agent RL. While feasibility of the results is limited to cases where prior knowledge allows strategy-wise decomposition, it is nonetheless an interesting step in this field. Reviewers are concerned that the above substantial limitation of the work has not been sufficiently discussed in the paper, and the authors are asked to clarify this aspect in a subsequent revision.

**Award:**

No

---

### Decision · Program_Chairs · 2022-09-14

Accept